# Quantification of a shelter cat population: Trends in intake, length of stay and outcome data of cats in seven Dutch shelters between 2006 and 2021

W. J. R. van der Leij[1]*, J. C. M. Vernooij[2], C. M. Vinke[3], R. J. Corbee[4], J. W. Hesselink[5]

1 Department of Clinical Sciences, Shelter Medicine Program, Faculty of Veterinary Medicine, Utrecht University, Utrecht, The Netherlands, 2 Department Population Health Sciences, Division Farm Animal Health, Faculty of Veterinary Medicine, Utrecht University, Utrecht, the Netherlands, 3 Department Population Health Sciences, Division Animals in Science & Society, Animal Behaviour, Faculty of Veterinary Medicine, Utrecht University, Utrecht, the Netherlands, 4 Department of Clinical Sciences, Clinical Nutrition, Faculty of Veterinary Medicine, Utrecht University, Utrecht, The Netherlands, 5 Department of Clinical Sciences, Faculty of Veterinary Medicine, Utrecht University, Utrecht, The Netherlands

* W.J.R.vanderleij@uu.nl

**Data Availability Statement:** the data are held in a public repository of DATAVERSE at: https://

## Abstract

Shelter metrics can be used by shelters for self-assessment to optimise the health of their animal population and to identify risk factors for disease outbreaks. However, there is a need for a wider scope of these shelter metrics, as evidenced by the interest from shelters in the benchmarking of shelter progress and the development of national best practices. For the first time, Dutch shelter data were used retrospectively to signal trends using potentially reliable metrics for the analysis of shelter data. The aims of this study were to apply relevant metrics describing the different phases of shelter management for shelter cats (i.e., intake, stay and outcome) and a retrospective analysis of shelter data over the period between 2006 and 2021. Seven of the approximately 120 Dutch animal shelters participated in this study. Quantitative data on the intake of more than 74,000 shelter cats (e.g., stray cats, cats surrendered by their owners and cats obtained from other sources) and their outcomes (i.e., cats rehomed, returned to their owners, deceased, or otherwise lost) have been analysed. Metrics such as rehoming rate, return to owner rate, rates for mortality and euthanasia, length of stay and risk-based live release rate were determined. The main findings of the study during this 16-year period were that, over time, the number of cats per 1000 residents admitted to Dutch shelters was reduced by 39%, the number of feline euthanasia cases decreased by approximately 50%, the length of stay showed a reducing trend, while the return to owner and the risk-based live release rate increased. The shelter metrics examined in this study could be helpful in monitoring and evaluating the management, consequent health, and well-being of cats in shelters and eventually measuring progress of shelters both in the Netherlands and at a European level.

dataverse.nl/dataset.xhtml?persistentId=doi:10.34894/WHA5IS.

**Funding:** W.J.R. van der Leij received financial support for this study from the DierenLot Foundation. Grant number: 2022-001 Funder: DierenLot Foundation URL of DierenLot Foundation: https://www.dier.nu/ The funder had no role in study design, data collection and analysis, decision to publish, or preparation of the manuscript. We are grateful to the Foundation for their helpful financial support of the Shelter Medicine program, which enabled us to conduct this research.

**Competing interests:** The authors have declared that no competing interests exist.

## Introduction

The domestic cat (*Felis silvestris catus*) is the most popular mammalian pet species in the Netherlands, with a total of almost 3 million individuals. One or more cats live in about a quarter of the approximately 8 million Dutch households [1]. Cats form the largest animal population in Dutch animal shelters: the animal population in a typical Dutch shelter consists of approximately 70% cats, 20% dogs and 10% other companion animal species [2]. When a stray animal is taken in by an animal shelter, a legally required holding period of 14 days applies [3]. The vaccination status of stray animals is unknown upon arrival at a shelter. Dutch law therefore requires that every cat be vaccinated against panleukopenia, feline herpes virus and feline calicivirus within 5 working days of arrival. During the 14-day holding period, the animals are solitary housed in quarantine units, while animals with clinical signs of disease are to isolation wards [4]. A pet surrendered by owners with a known vaccination status can in some cases be rehomed immediately. Dutch law only allows euthanasia of dogs and cats if the animal poses an immediate danger to humans or animals, when a veterinarian has determined that euthanasia is in the best interest of the animal, or to end the unbearable suffering of the animal. Therefore, Dutch shelters do not facilitate owners' requests for euthanasia of their pet, as this assessment is invariably made by the shelter in consultation between employees, shelter veterinarians and other people with relevant expertise, based on the animal's health and well-being. Only veterinarians are legally allowed to perform euthanasia [4, 5].

Approximately 120 open admission shelters with municipal contracts [6] provide care for stray and owner surrendered cats in the Netherlands and strive to optimise their welfare during the shelter stay with the fastest possible live outcome for the animals [7]. After entering a shelter, an animal will go through three different stages: the intake, the stay (including foster care), and the outcome. Specific metrics have been used for each phase to indicate the performance of a shelter [8, 9]. Some of these performance measures are the number of cats entering per unit of time, the number of care days and the outcome per cat. These factors could significantly affect the welfare of shelter cats because overcrowding and extended length of stay increase the risks of stress and diseases, resulting in reduced animal welfare [10, 11].

Shelter metrics are essential for the self-assessment of shelters and Dutch animal shelters keep records of the animals in their care, as legally required by stakeholders such as municipalities. However, there is also a need for wider use of these metrics to benchmark the progress of shelters [8, 12]. When these data are not restricted to individual shelters but shared with the shelter community, trends in the pet population and its effect on shelters can be monitored on national and possibly international levels. Weiss [13] showed that when shelter organisations collaborate in collecting accurate data based on well-defined definitions, the percentage of animals leaving a shelter alive improved, regardless of the differences in the intake between the cooperating shelters.

There are no collective Dutch annual shelter data currently available. Given that cats are the largest group in shelters, we chose to analyse feline data in this study of shelters from 2006 to 2021, to gain retrospective insight into the current Dutch shelter situation. The aim of this research is twofold: first, to introduce key metrics as indicators of shelter performance in the different phases of feline shelter care (intake, stay (including foster care) and outcome) for shelters in the Netherlands and, second, to identify trends in these metrics during the years 2006 to 2021.

## Materials and methods

### Selection of animal shelters

A convenience sample of seven local shelters, willing to participate in this study under the condition of anonymity, provided annual intake and outcome data for approximately 74,000

individual cats entering and leaving the shelters from 2006 through 2021. All shelters provided care for cats as well as dogs, while two shelters also housed other pet species such as rabbits, rodents, and pet birds. Being located in four different Dutch provinces, the shelters differed geographically, while the shelter service areas (the local communities served by these shelters) ranged from hardly urbanised to extremely urbanised and the averaged annual income per resident in the shelter care area varied between € 24,000 and € 29,300 (S1 Table).

## Description of shelter metrics

The key shelter metrics in this study were intake per source, outcome, and length of stay (LOS). The definitions of these metrics are presented in Table 1.

The metrics were compatible with the two shelter software programs used (except for one variable, described later) for a reliable analysis of the data. Rates were calculated from the data

**Table 1. Overview of the shelter metrics used in this study with definitions and abbreviations\*.**

| Metrics (abbrevation) | Definition |
|---|---|
| | **Phase: INTAKE** |
| Total Annual Intake[a] | Total number of cats entering the care of an animal shelter each year (including cats for foster care). |
| Stray Cat (SC)[b] | Cat which left or lost its home or was abandoned by its legal owner. |
| Owner Surrendered Cat (OSC) | Privately owned cat for which responsibility and care has been transferred to an animal shelter by the previous owner, who therewith terminates ownership of the pet. |
| Returned Adoptions (RetA) | Rehomed shelter cats returned to the shelter within a specified time frame by their new owners after adoption. |
| | Additional information: shelters used different time frames for this parameter, making comparison over time difficult. This parameter was therefore only used in the overview of the total annual intake of shelter cats. |
| Others | Cats from different sources: Trap–Neuter and Release programs (TNR); shelter-born kittens; cats taken in from other shelters ('transfers-in'); residuals. |
| | **Phase: OUTCOME** |
| Outcome | Final result of a stay in a shelter when the cat leaves the shelter (dead or alive). |
| Rehomed | Shelter cat adopted by new owners. |
| Returned To Owner (RTO)[c] | Stray cat (SC) successfully reunited with its rightful owners. |
| Euthanasia | Induced death of shelter cat when welfare cannot be managed, due to physical, behavioural, or environmental conditions. |
| Mortality[a] | Shelter cat deceased by natural cause or by euthanasia. |
| Other | Shelter cat leaving shelter alive for other reasons, for example exchange of cats with other shelters or returned to field. |
| Length Of Stay (LOS) | Number of days between date of intake and date of outcome (= departure from the shelter) + 1 day. |
| | Additional information: the one additional day counts for the time/care this animal had received in the shelter if it returned to its owner on the day of intake (which would result in a LOS of 0 days). |
| Still In Shelter (SIS)[a] | Cats still in shelter (SIS) on 31st of December of a particular year, without outcome yet (but using care and housing of the shelter while waiting for an outcome). |

\*Remarks and abbreviations. SC = Cats of all ages were included in this category and a shelter holding period of 2 weeks is mandatory for cats in this category. OSC = for cats from this category a holding period is not mandatory but depends on the medical history of an individual animal. LOS = the LOS is calculated per year per shelter by taking the median of the days of all individual cats with an outcome. When the 'mean of the LOS' is mentioned in the results, this is the 'mean of median LOS'. To avoid confusion, we prefer to write the 'mean LOS' over the 'mean of median LOS'. The SIS of this study is measured annually on the 31st of December.
Table footnotes [a] [14], [b] [15], [c] [13].

metrics, such as rehoming rate (RR), return to owner (RTO) and risk-based live release rate (RLRR). The rates are presented in Table 2.

The individual cat data were summarised per shelter as total number per year, total number per 1000 residents and per cat source. Outcomes for shelter cats could be twofold: a live release (including rehoming, return to owner (RTO) or other reasons for a cat to leave the shelter alive) or a non-live outcome in which the cat was euthanised or found dead. The length of stay (LOS) of a feline shelter population is calculated per year per shelter by taking the median of the days of all individual cats with an outcome. When the 'mean of the LOS' is mentioned in the results, this is the 'mean of median LOS'. To avoid confusion, we prefer to write the 'mean LOS' rather than the 'mean of median LOS'.

The smallest shelter (S1 Fig, darkest blue line) in this study showed an irregular annual LOS (S5 Fig) and risk-based live release rate (RLRR) (S6 Fig). Data from this shelter have therefore been excluded from the subsequent analysis of LOS and RLRR.

Euthanasia at the owner's request does not play a role in this study, because this practice is unusual for Dutch animal shelters.

**Table 2. Overview of the metric rates calculated in this study with definitions and abbreviations.**

| Rates and abbrevations | Formula + additional information |
|---|---|
| **Risk-based Live Release Rate (RLRR)** | = (# cats leaving the shelter alive) / (total # cats with an outcome (dead or alive) + # SIS cats) × 100 |
| | Additional information: in the denominator of RLRR, cats are included that do not have an outcome yet (but are still in need of housing and care from the shelter while waiting for an outcome (= SIS-category)). RLRR reflects the probability of an individual cat in the shelter being released alive. As the RLRR includes the animals still in the shelter, this rate also signals trends in the shelter population such as overpopulation ('creeping intake' [7]). |
| **Rehoming Rate (RR)** | = (# cats adopted) / (total # cats with an outcome (dead or alive)) × 100 |
| **Return To Owner (RTO)** | = (# SC who are reunited with their rightful owners) / (total # SC taken in) × 100 |
| **Mortality Rate (MR)** | = (# cats euthanised or found dead) / (total # cats in shelter (# new cats + # SIS cats) × 100 |
| | Additional information: MR represents the non-live outcome as part of the total annual shelter population, counting all cats with and without an Outcome (= SIS-category). Since it represents the total shelter cat population (not only for cats with an outcome) it also reflects the chance of an individual cat dying in a shelter either from human intervention or natural death, and according to Scarlett [14] is the proper way to calculate the total non-live outcome. |
| **Euthanasia Rate (ER)** | = (# cats euthanised) / (total # cats in shelter (# new cats + # SIS cats)) × 100 |
| | Additional information: ER represents the non-live outcome as part of the total annual shelter population, counting all cats with and without an Outcome (= SIS-category). Since it represents the total shelter cat population (not only cats with an outcome) it also reflects the chance of an individual cat being euthanised, and according to Scarlett [14] is the proper way to calculate the total euthanasia rate. |
| **Death Rate (DR)** | = (# cats euthanised or found dead) / (total # cats with an outcome (dead or alive)) × 100 |
| | Additional information: the denominator of DR does not include cats from the SIS category and gives the relationship between live and non-live outcomes of cats leaving the shelter. Although, according to Scarlett [14], MR and ER (as part of the total shelter population) are the proper way to calculate the non-live outcome, this study also reports DR as it is used in several European studies [16, 17]. Important: the DR is always slightly higher than the MR, as the DR-denominator is smaller because the SIS-numbers are excluded. |

*Remarks and abbreviations. For rates, the same period is used for the number of cats in the numerator as for the number of cats in the denominator [14].

# = number of.

## Standardising shelter data

Shelter service areas vary in their human population and, as a result, shelters differ in their capacity and metrics. Given these differences, shelter metrics were standardised per 1000 human inhabitants [18] in the service area of a shelter. For this, the officially registered annual number of residents per service area (= municipality) is used [19]. During the 16-year time span, municipal merging between neighbouring counties changed the number of residents per shelter service area. To integrate these municipal mutations into our analysis, the actual population size of the service area during the year of merging was used in the standardisation of the shelter data.

The inclusion criteria for shelters were the registration of shelter animals by using shelter software and the availability of multi-year intake and outcome data (preferably the years 2006 to 2021, inclusive). The seven participating shelters used Dutch commercial shelter software to register their animals: five of them used the program 'DIPO' [20] and two used the program 'DOCASOFT' [21].

All participating shelters in this study gave signed informed consent.

## Statistics

From seven shelters, data on the intake of 74,016 individual cats and on 74,129 cats with an outcome were loaded into Microsoft Excel spreadsheets and the metrics were calculated and standardised per 1000 residents according to the formulas in Table 2. Calculated metrics were summarised in yearly intervals and visualised by line plots over the course of time. Regular trendlines were added to Figs 1A–11A, only to visualise the trends per shelter; these lines were not used for statistical analysis. When mean values of metrics are given, the standard deviation (SD) is given as well.

Data supplied by the smallest shelter in this study regarding the LOS and the RLRR from 2020 and 2021 deviated considerably (see S5 and S6 Figs, darkest blue line) from the metrics of the other shelters. Owing to its limited size, alterations in the regular shelter management (for

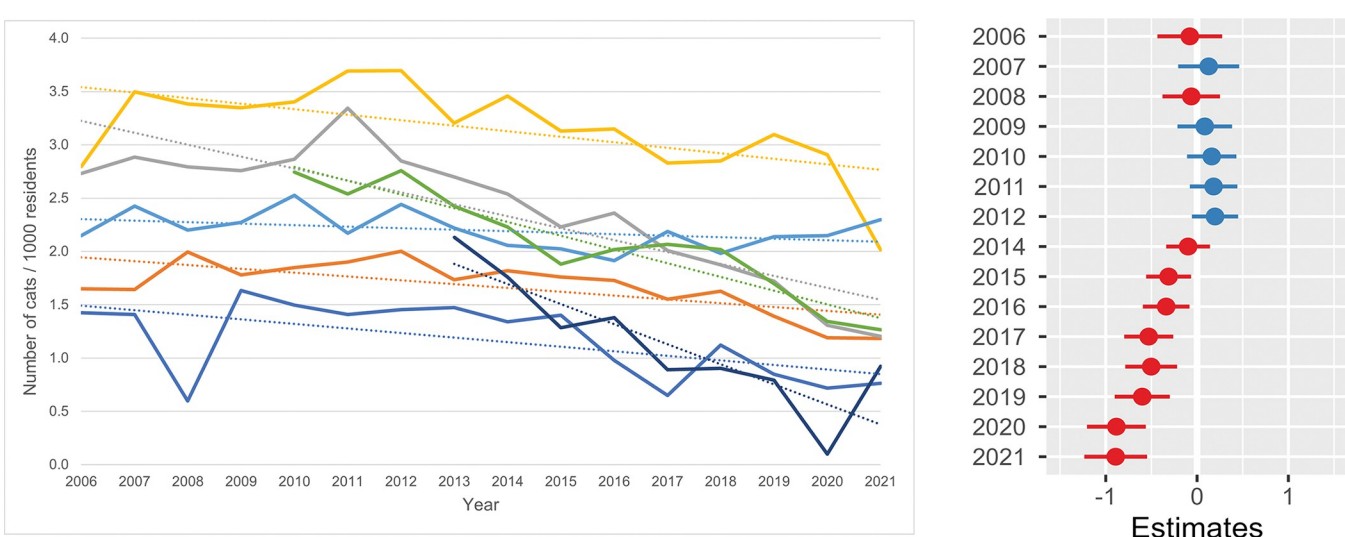

**Fig 1. a. Annual total intake of cats / 1000 residents / shelter.** The annual feline intake data per 1000 residents in the shelter service area were used. One shelter had missing information between 2006 and 2009 and another shelter between 2006 and 2012. All seven shelters were included in the data for the years 2013 through 2021. **b. Estimates for the average difference in total annual intake / 1000 residents.** Estimates for the average difference in total annual intake per 1000 residents compared with the annual intake in 2013 in seven shelters. The horizontal lines represent the estimate (the coloured dot) with 95% confidence intervals. An estimate of 0 (zero) means no difference from the total annual intake in 2013.

example during the SARS-CoV2 pandemic in 2020–2022) may have had more impact on its metrics compared with the larger shelters. We therefore excluded the LOS and RLRR data of this small shelter from the general analysis.

All metrics per year, per shelter and per 1000 residents were transferred to the statistical program R version 4.0.5 [22] for statistical analysis. For each shelter, the metrics were standardized and calculated per year. The metrics were repeatedly calculated and therefore correlated within the shelter. Although the metrics were standardized, the level of the metrics i.e., "CATS IN per year" for a specific shelter can be affected by its shelter management, finances etc. Simple linear regression models are based on linearity between outcome variable and continuous independent variable, normal distribution, and homoscedasticity (constant variability) of the residuals of independent observations. When the independent variable is categorical then this is called ANOVA and belongs to the family of linear regression models. The assumption of independence is violated in our data collection and the dependency between measurements of the same metric within shelter should be incorporated in the statistical model. We therefore used a linear mixed effects model which is a combination of estimation of fixed effects and so-called random effects i.e., estimation of variability between the independent subjects (shelter) as metrics were measured yearly within the same shelter. For comparison a very simple example of a model taking dependency of measurements into account is the paired t-test with two conditions within the same subject (i.e., body weight before and after diet treatment of the same person).

Linear mixed effects regression analysis [23] was applied for each standardised metric as the outcome variable, with year as an explanatory factor. A random effect for shelter was added to the model to account for repeated measurements within a shelter. Year 2013 was taken as the reference year because all metrics were available from this year onwards for all shelters. Each metric was calculated per year and shelter and was visualised in line plots. The model estimates with 95% confidence intervals were presented in forest plots [24]. The estimates of the models should be interpreted as the difference between the mean number for a specific year compared with the mean number in 2013. The residuals of each of the models (S1 File) were used to check model validity for normality and homoscedasticity (constancy of variance) and no serious aberration was observed.

The metrics 'Euthanasia/1000 residents' and 'LOS CATS Total Median' were log transformed to meet the model assumptions. The resulting estimates of the log transformed models should be interpreted as a ratio: i.e., an estimate of 0.9 means that the mean number in the specific year is 0.9 times as large (e.g., 10% lower) as the mean number in the reference year 2013.

## Results

The quantitative data on the intake of 74,016 shelter cats were analysed. The intake of cats was categorised into stray cats (SC), owner surrendered cats (OSC), cats returned to the shelter after adoptions (RetA) and cats coming from other sources (including trap–neuter–return programs (TNR), exchange with other shelters (= 'transfers-in') and born in the shelter). The outcome for shelter cats was specified as rehomed, returned to owners (RTO), non-live outcome (mortality: natural death plus euthanasia) and other outcomes (= exchange with other shelters or cats leaving the shelter alive for other reasons such as return to field). From these data several rates were determined, including rehoming rate (RR), return to owner rate (RTO), non-live outcome rates such as mortality rate (MR), death rate (DR) and euthanasia rate (ER), risk-based live release rate (RLRR) and the length of stay (LOS). In the results described below, the data and trends of the intake are presented, followed by those of the stay and outcome.

## Intake

The total intake differed considerable among the seven shelters: the two largest shelters received more than 800 cats annually, three shelters of medium size had an intake of between 400 and 800 cats per year, while the two smallest shelters in this study had intakes of fewer than 400 cats annually (S1 and S2 Figs). Regardless of shelter size, the total intake of cats in absolute numbers decreased between 2006 and 2021 for all shelters in this study.

The total annual standardised intake of cats per 1000 residents also decreased with time (Fig 1A). A total intake of 1.43–2.80 cats/1000 residents in 2006 dropped to 0.76–2.30 cats per 1000 inhabitants in the pandemic year 2021. The result of the estimated average difference (i.e., difference between means) in the total annual intake per 1000 residents of the seven shelters compared with the annual intake in year 2013 is shown in Fig 1B (S2 Table), with 2013 taken as the reference year. Between 2006 and 2013 the annual total intake of cats per 1000 residents did not change drastically (all confidence intervals of the annual intakes included zero = no difference), but after 2013 a clear decrease was observed: on average one fewer cat per 1000 residents was taken into shelter care in 2021 compared with 2013, a reduction of 39%.

## Source

The 74,016 cats came from different sources: SC, OSC, RetA and cats entering for other reasons (kittens born in the shelter, transfers-in, etc.). Fig 2 shows the sources per year from which the cats came.

Stray cats were by far the major source of cats in shelters: the average annual intake of SC in this study was 74.1% (SD = 2.2, range: 71.5% in 2012/2015 to 80.2% in 2021) of the total intake during the 16 years. The second category were the OSC, accounting for 13.7% (SD = 1.8) of all cats. The RetA category accounted for 6.5% (SD = 0.7). The 'Others' category of 5.7% (SD = 1.1) included cats from TNR programs (0.7%), shelter-born kittens (3.2%), transfer-ins (0.4%) and a residual category of 1.3%.

In Fig 3A the annual intake of SC per 1000 residents is shown, while the estimated difference in the annual intake of SC per 1000 residents compared with the annual intake in 2013 is presented in Fig 3B (S2 Table). The SC intake showed the same trend as the total annual feline intake: between 2006 and 2013 no systematic change was observed, but from 2013 the annual intake of SC showed a significant decrease of on average more than 0.5 cat/1000 residents, a reduction of 34%.

The second source of shelter cats was the owner-surrendered cat (OSC): in total 13.7% (SD = 1.8) of all shelter cats were surrendered by their owners/care givers. Five of the seven shelters showed decreasing trends for the OSC (Fig 4A). The OSC/1000 residents varied between 0.23 and 0.34 in 2006 and dropped to 0.06–0.26 in 2019. Eventually, in 2021, the OSC/1000 residents decreased to 0.0–0.31. The estimated difference in the average number of OSC/1000 residents of each year compared with 2013 is presented in Fig 4B (S2 Table). During the years preceding 2013 the OSC/1000 residents remained stable but from 2013 the OSC/1000 residents decreased quite linearly to almost 0.25 fewer cats in 2021.

The third source, RetA, accounted for 6.6% of the total intake (4866/74,016). However, these data have not been used for analysis because of differing definitions among shelters, such as the period after which a returned cat should be registered as 'returned after adoption'.

## Outcome

**Annual outcome.** The annual distribution of the outcome categories is pictured in Fig 5. Most shelter cats were rehomed, with a total average RR of 67.8% (SD = 2.4). The second most

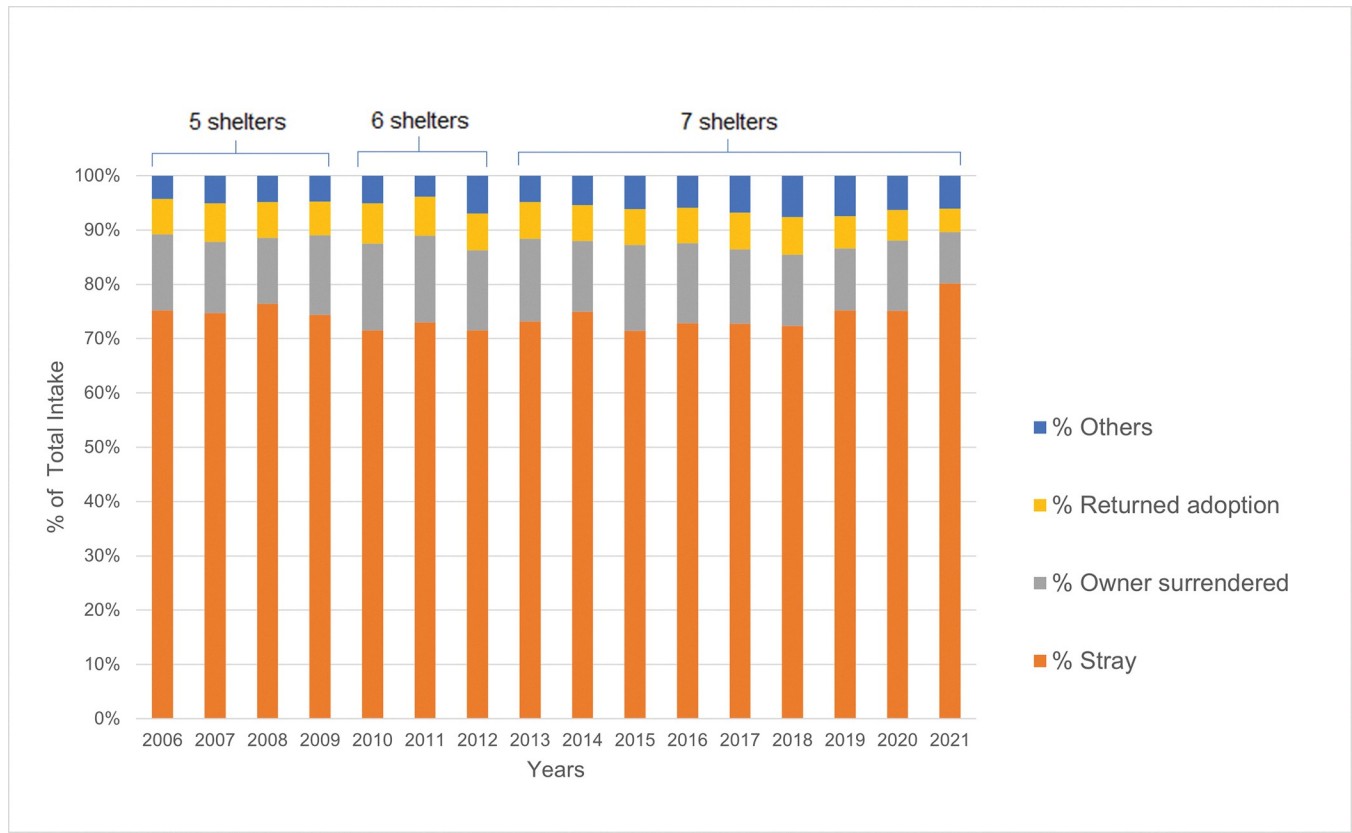

**Fig 2. Annual distribution sources of intake of shelter cats.** Annual distribution of sources of intake of cats of the seven participating shelters combined. One shelter had missing information between 2006 and 2009 and another shelter between 2006 and 2012. All seven shelters were included in the data for the years 2013 through 2021. The annual intake was subdivided into different sources of cats: stray cats, owner surrendered, returned adoptions and other sources. In total, data from 74,016 cats were used.

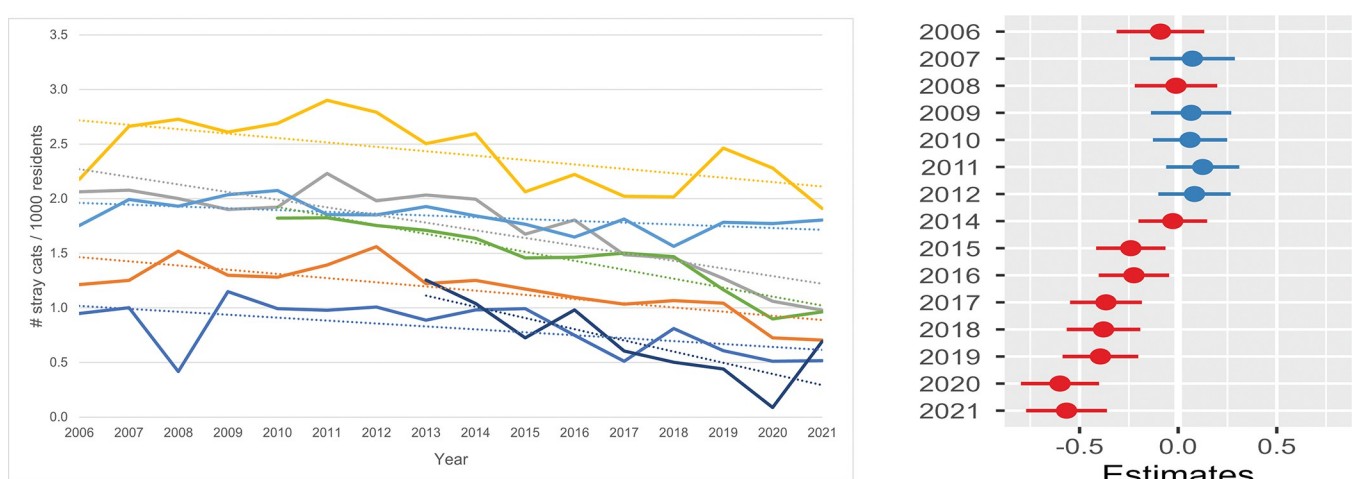

**Fig 3. Annual intake stray cats / 1000 residents / shelter. a.** The annual feline intake data for stray cats per 1000 residents in the shelter service area is used. One shelter had missing information between 2006 and 2009 and another shelter between 2006 and 2012. All seven shelters were included in the data for the years 2013 through 2021. **b. Estimates for the average difference in annual intake of stray cats / 1000 residents.** Estimates for the average difference in annual intake of stray cats per 1000 residents compared with the annual intake in 2013 for seven shelters. The horizontal lines represent the estimate (the coloured dot) with 95% confidence intervals. An estimate of 0 (zero) means no difference from the annual intake of stray cats in 2013.

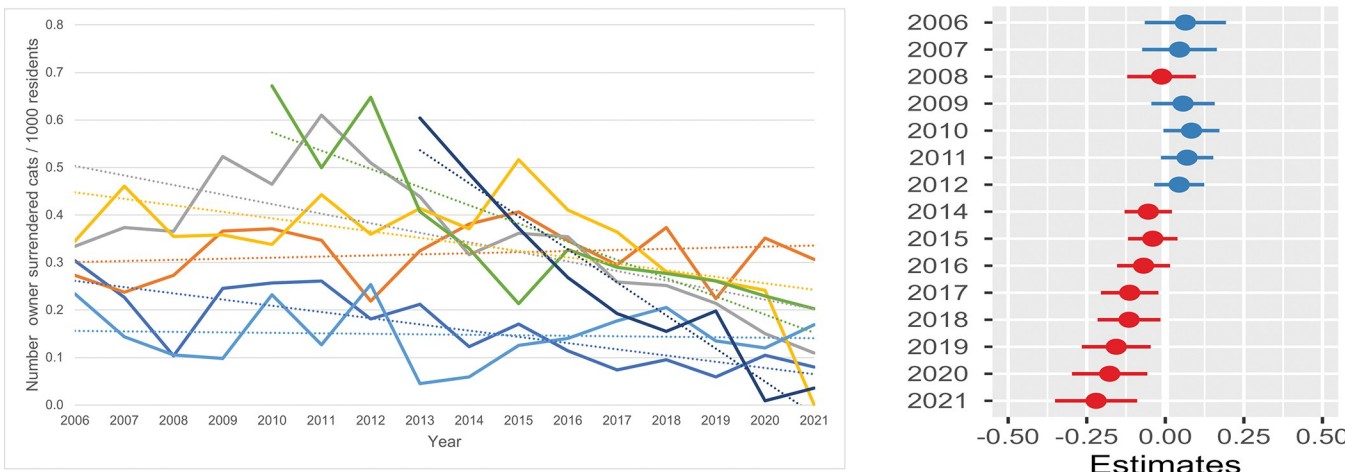

**Fig 4. Annual intake owner surrendered cats (OSC) / 1000 residents / shelter. a.** The annual feline intake data per 1000 residents in the shelter service area were used. One shelter had missing information between 2006 and 2009 and another shelter between 2006 and 2012. All seven shelters were included in the data for the years 2013 through 2021. **b. Estimates for the average difference in annual intake of owner surrendered cats (OSC) / 1000 residents.** Estimates for the average difference in annual intake of owner surrendered cats (OSC) per 1000 residents compared with the annual intake in 2013 for seven shelters. The horizontal lines represent the estimate (the coloured dot) with 95% confidence intervals. An estimate of 0 (zero) means no difference from the annual intake of OSC in 2013.

common outcome for cats was being returned to their owners (RTO): an annual average of 11.6% (SD = 2.0). Some 4.8% (SD = 1.6) of the cats left the shelter alive for other reasons. For 8.8% (SD = 1.3) of the feline population the non-live outcome was euthanasia or natural death (= mortality rate (MR)). The population Still In the Shelter (SIS-category) consisted of 7.1% (SD = 1.7) of all shelter cats counted on the 31st of December each year. During the pandemic year 2020 the SIS dropped to a minimum of less than 4% while spiking the next year in the pandemic, in 2021, to more than 10%.

Per individual shelter the annual outcomes differed. Six of the seven shelters showed a decreasing rehoming rate (RR) during the 16 years (Fig 6A). In one shelter the most prominent decrease was seen when its RR reduced from an average of 67% (SD = 4.6) during the years 2006–2013 to 58% (SD = 4.5) during 2014–2021. Only the smallest shelter (dark blue line) showed an increase in the RR. The results from the statistical model (Fig 6B and S3 Table) showed an unstable trend from 2013 onwards and no systematic change could be observed, while in 2010 the RR was 5% higher than in 2013.

The second outcome for cats in shelters is being returned to their owners (RTO). To evaluate the efficacy of this process in the participating shelters, the RTO is presented as a rate of the annual intake numbers of SC per shelter (Fig 7A). Six shelters showed an increase in the RTO over time. The smallest shelter showed a decreasing RTO. One shelter with an average RTO of 26% (SD = 3.1) during the years 2006–2013 increased its RTO to 45% (SD = 4.6) in 2014–2021. The estimated difference in the RTO over time of all participating shelters is presented in Fig 7B (and S3 Table). During the years 2006–2013, the RTO did not change significantly as all confidence intervals covered the zero value. From 2013 onwards, all estimates are positive but still covering zero. Therefore, the chances of a shelter cat being successfully reunited with its owners increased slightly in all shelters to between 19 and 23% (means of seven shelters 2014–2021) but remained quite variable among shelters over the course of time.

The third outcome of live release (average 5.1%, SD = 1.8) represents cats leaving the shelter alive for other reasons. These reasons can be very diverse such as transfer of cats to other rescue centres or in a few individual cases community/feral cats who were neutered and returned to the field.

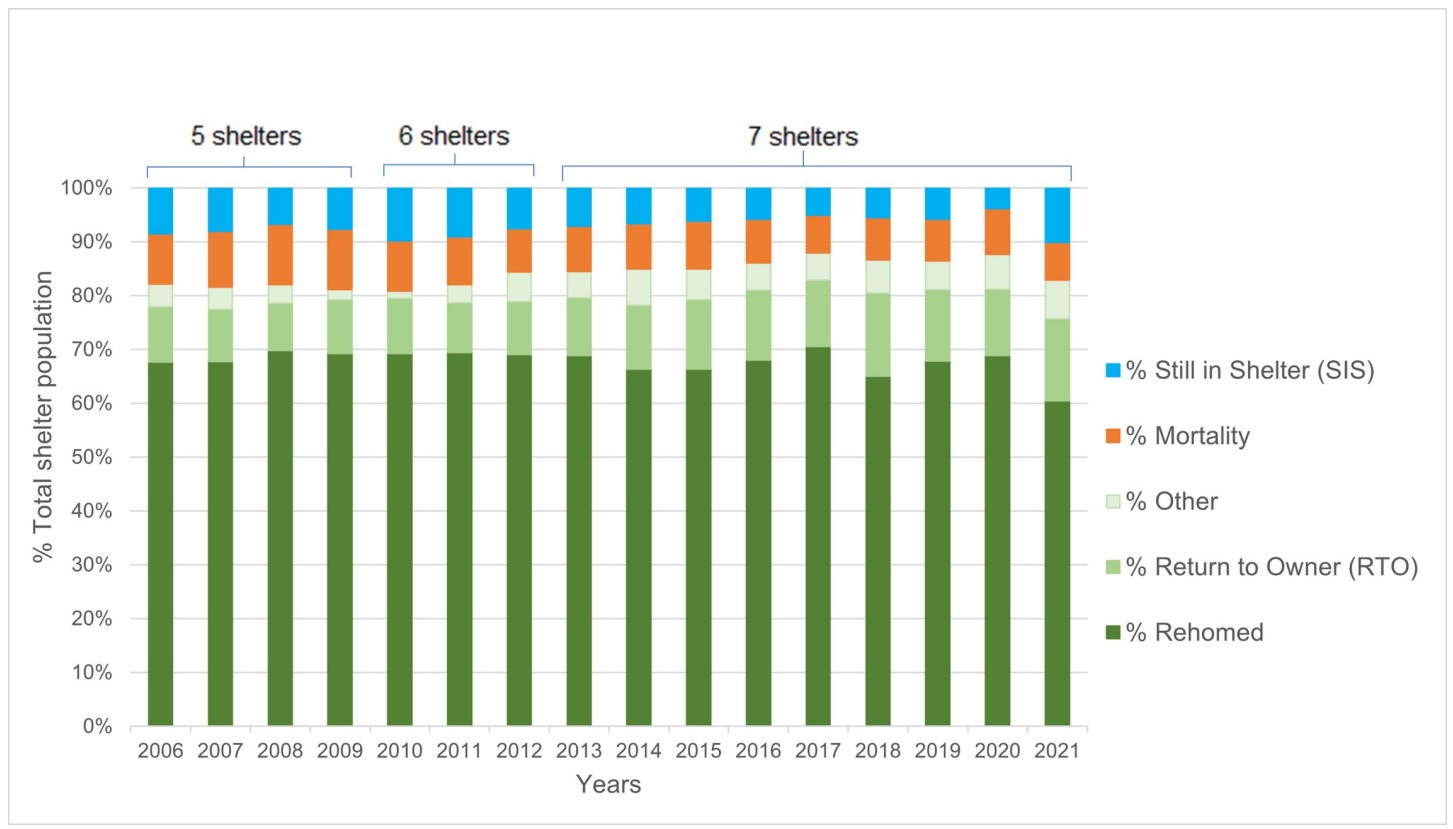

**Fig 5. Annual distribution of categories of shelter cats.** Annual distribution of categories of Outcome and Still in Shelter for shelter cats of seven shelters combined. The annual outcome has two categories: live release and non-live release of cats. The live release consists of 'Rehomed', 'Returned to Owner (RTO)' and 'Other'. The non-live release is 'Mortality' (consisting of the number of feline euthanasia and found dead cases). The cats 'Still in Shelter' are without outcome on the 31st of December of each year. One shelter had missing information between 2006 and 2009 and another shelter between 2006 and 2012. All seven shelters were included in the data for the years 2013 through 2021. In total, data on 74,129 cats were used.

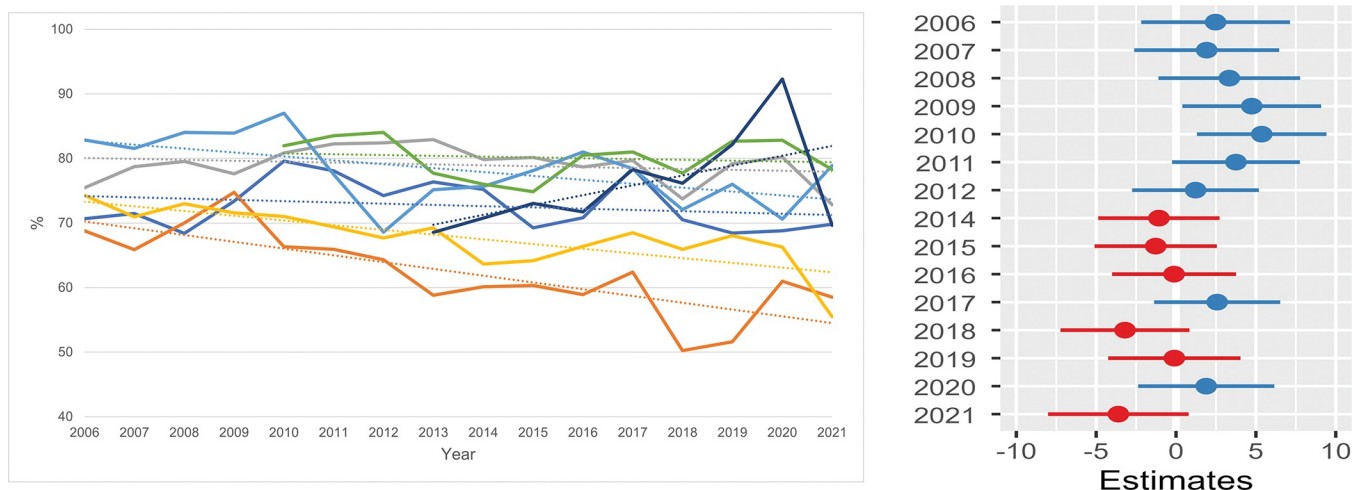

**Fig 6. Rehoming rate (RR) / shelter. a.** The annual feline RR (% rehoming of total outcome) per shelter is shown. One shelter had missing information between 2006 and 2009 and another shelter between 2006 and 2012. All seven shelters were included in the data for the years 2013 through 2021. **b. Estimates for the average difference in rehoming rate (RR).** Estimates for the average difference in RR compared with the RR in seven shelters in 2013. The horizontal lines represent the estimate (the coloured dot) with 95% confidence intervals. An estimate of 0 (zero) means no difference from the annual RR in 2013.

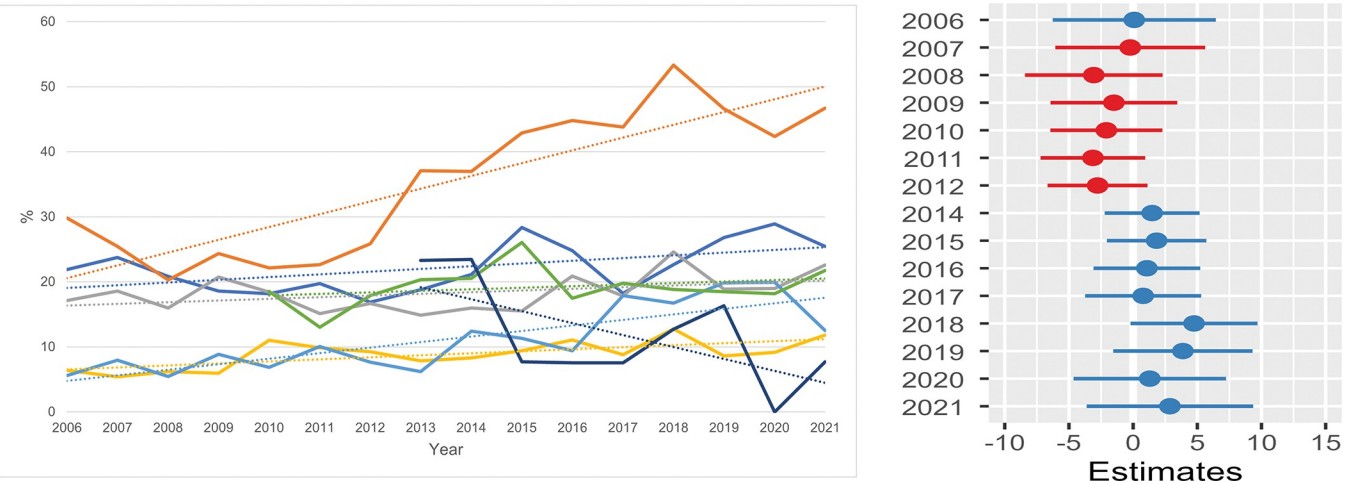

**Fig 7. Stray cats returned to owner (RTO) / shelter. a.** The annual feline RTO-rates (% of stray cats returned to their owner) per shelter are shown. One shelter had missing information between 2006 and 2009 and another shelter between 2006 and 2012. All seven shelters were included in the data for the years 2013 through 2021. **b. Estimates for the average difference in return to owner (RTO).** Estimates for the average difference in return to owner (RTO) from the RTO for seven shelters in 2013. The horizontal lines represent the estimate (the coloured dot) with 95% confidence intervals. An estimate of 0 (zero) means no difference from the annual RTO in 2013.

**Euthanasia rate (ER).** The average ER of the total population of shelter cats was 7.2% (SD = 1.3, range: 9.5% in 2008 to 5.5% in 2021). The shelter euthanasia numbers per 1000 human residents (Fig 8A) decreased in those 16 years. In 2006 between 0.08 and 0.45 cats per 1000 residents were euthanised, which decreased in 2019 to 0.03–0.31 cats/1000 residents and dropped to 0.03–0.16 cats per 1000 residents in 2021. The results from the statistical model (see Fig 8B and S3 Table) showed rather stable ERs till 2015 and the reduction was most prominent in the last 3 years, to around 50% of the ER level in 2013.

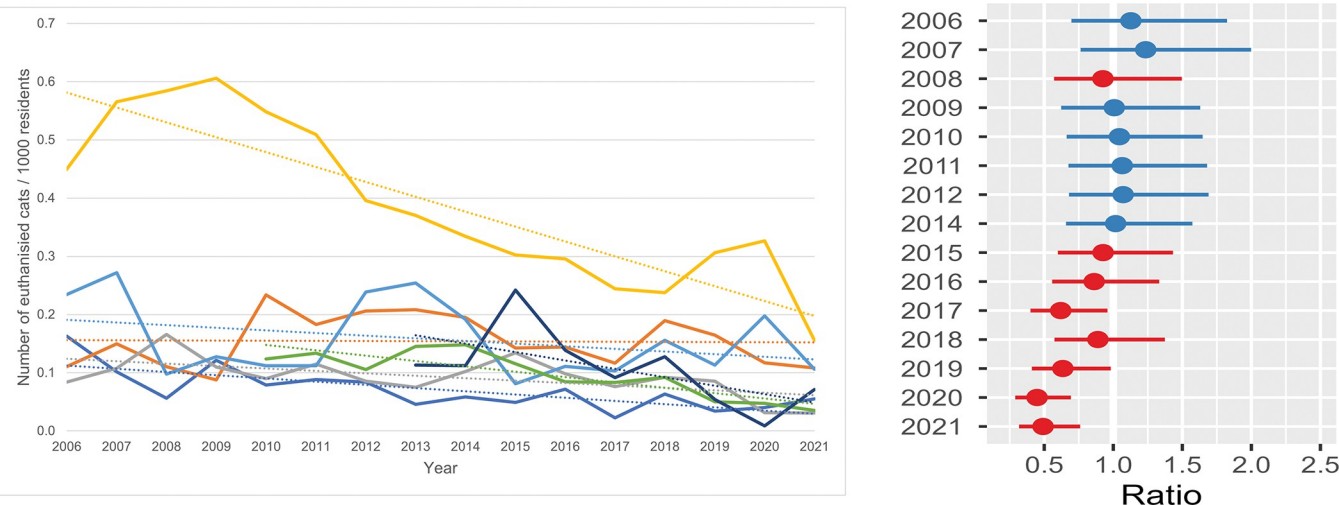

**Fig 8. Euthanasia / 1000 residents / shelter. a.** The annual feline cases of euthanasia per 1000 residents in the shelter service area is pictured. One shelter had missing information between 2006 and 2009 and another shelter between 2006 and 2012. All seven shelters were included in the data for the years 2013 through 2021. **b. The ratio of the mean difference in annual euthanasia cases / 1000 residents.** The log transformed estimated ratio of the mean difference in annual euthanasia cases (from seven shelters) per 1000 residents compared to the annual cases in the reference year 2013. For 'Euthanasia/1000 residents' the metric was log transformed to meet the model assumptions. The horizontal lines represent the estimate (the coloured dot) with 95% confidence intervals. A ratio of 1 means no difference from the annual number of euthanasia cases in 2013.

**Death rate (DR).**   The DR is the number of deceased cats (by euthanasia or natural death) out of the total feline outcome (but excluding the SIS-category). When calculated over the complete 16-year period, 9.5% (SD = 4.2) of all cats recorded during that time did not leave the shelter alive. The DR proved to be quite variable in the course of time within and between shelters (S3 Fig). After an initial increase the average DR stabilised from around 2009 to 2016 and showed a slight decrease from then on (S4 Fig).

**Length of stay (LOS).**   The mean LOS for all cats in the seven shelters during the total 16 years was 26 days. S5 Fig pictures the annual LOS of the seven shelters in this study. The smallest shelter in this study has an irregular LOS (S5 Fig–dark blue line) and is therefore excluded from the LOS analysis. The source of shelter cats has influence on the LOS. Small kittens entering/being born in a shelter have the longest LOS: the median was 64 days (mean = 67.6 days, SD = 20.3). The median stay of all SC (including the RTO) was 25 days (mean: 26.0 days, SD = 10.6), while OSCs had a median of 27 days (mean = 28.6 days, SD = 15.1). SCs that were successfully reunited with their owners had a median shelter stay of 3 days (mean = 3.0 days, SD = 1.1).

The mean LOS of its total feline population varied considerably between and within the six shelters: the mean annual LOS of one shelter ranged between 16.0 and 25.0 days in the study period while another showed the highest annual mean LOS of between 23.0 to 39.0 days. Four shelters showed a decreasing trend while two shelters had an increase in the LOS (Fig 9A). Fig 9B (S4 Table) showed an initial increase from 2006 to 2011. From 2013 on the LOS seemed to reduce by 5 to 10%, but this was not statistically significant (except for 2018).

The LOS of all SC (Fig 10A, including SC returned to their owners (RTO)) is less variable among shelters compared with the LOS of all cats shown in Fig 9A. Also, less variability is observed within shelters, although one shelter showed a prominent change from 20–25 days to 5–10 days in later years. On average the years before 2013 showed rather stable mean LOS levels (Fig 10B, S4 Table), while from 2014 onwards the LOS decreased by even 7 days in 2018 and a slight increase during the years of the pandemic.

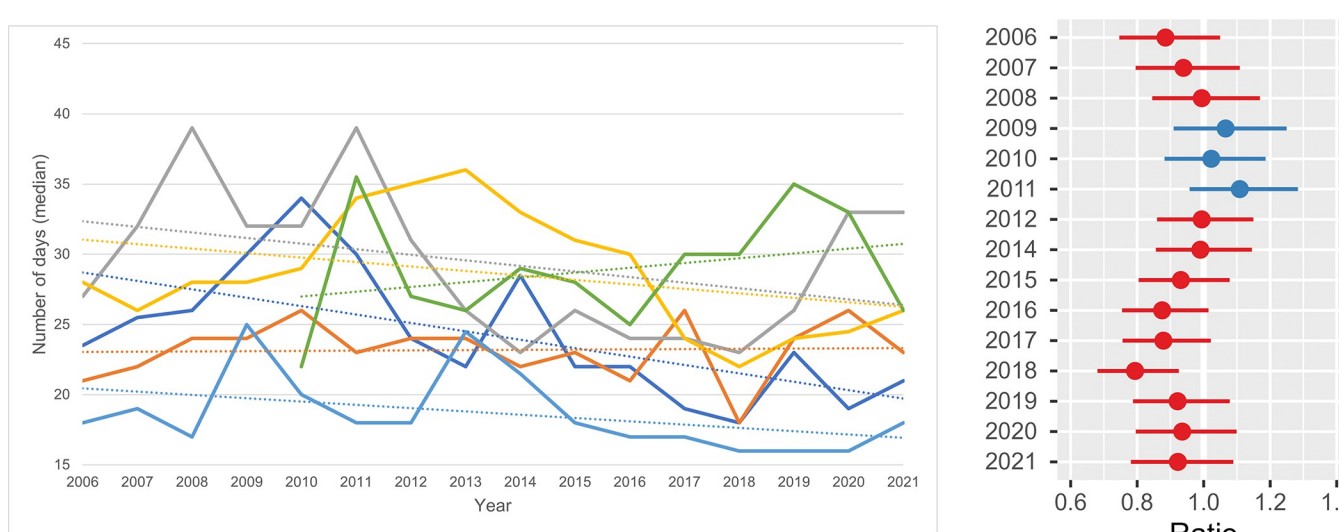

**Fig 9. Length of stay (LOS) in total / shelter. a.** The annual LOS of all cats is shown per shelter. One shelter had missing information between 2006 and 2009. Six shelters were included in the data for the years 2013 through 2021 (shelter G was not included). **b. The estimated ratio of the mean length of stay (LOS).** The log transformed estimated ratio of the mean LOS in each year against the mean LOS in the reference year (2013) of six shelters. For 'LOS Total/shelter' the metric was log transformed to meet the model assumptions. The horizontal lines represent the estimate (the coloured dot) with 95% confidence intervals. A ratio of 1 means no difference from the annual LOS in 2013.

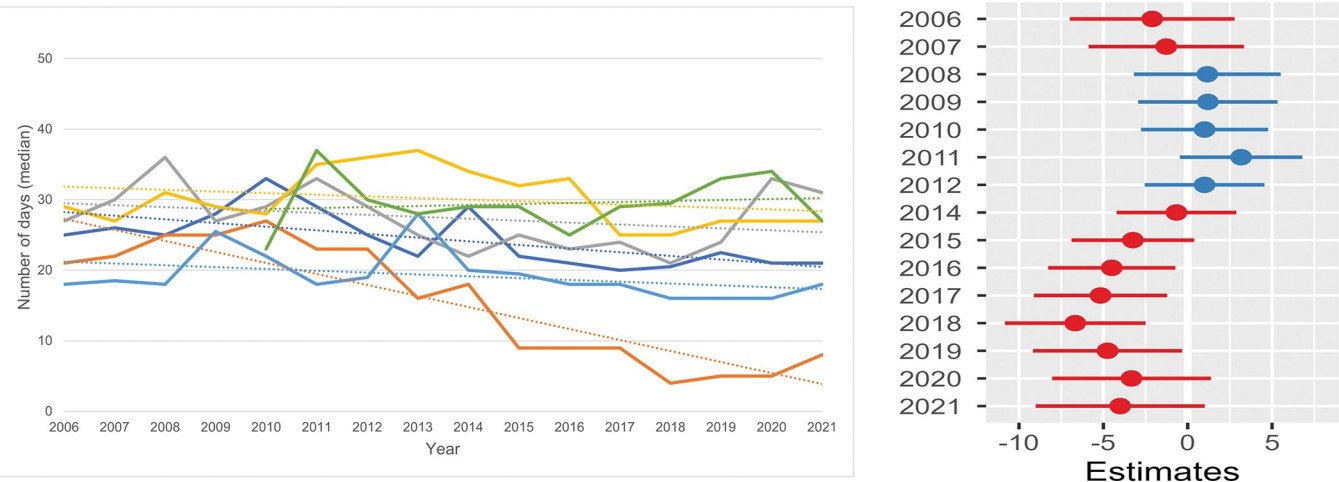

**Fig 10. Length of stay (LOS) of stray cats (SC) / shelter. a.** The annual LOS of all SC is shown per shelter and includes SC which are eventually returned to their owners (RTOs). One shelter had missing information between 2006 and 2009. Six shelters were included in the data for the years 2013 through 2021 (shelter G was not included). **b. The estimated difference in the mean length of stay (LOS) of stray cats (SC) / shelter.** The estimated difference in the mean length of stay (LOS) of stray cats (SC) per shelter of each year compared with the reference year (2013) in six shelters. The horizontal lines represent the estimate (the coloured dot) with 95% confidence intervals. An estimate of 0 (zero) means no difference from the annual LOS in 2013.

Data for shelter A (Fig 10A: darker blue line) and shelter B (orange line) show how an increase in the RTO affected the LOS of the SC. The total feline population of shelter A during these 16 years consisted of 70.4% SC. During this study the LOS of all SC (including the RTO-cats) decreased by 36% from 33 days in 2010 to 21 days in 2020 and 2021, while their RTO increased from 19% in 2006 to 25% in 2021. Shelter B, with a comparable feline population of 69.8% stray cats, had a gradual increase of the RTO from around 20% in 2008 to around 45% in 2019–2020. During the same time the LOS of SC (including the RTO-cats) was reduced by 80% from 25 days to 5 days in 2019 and 2020.

**Risk-based live release rate (RLRR).** The RLRR reflects the probability (= chance) of cats being released alive from the shelter during a certain time interval, as the denominator of the RLRR contains all cats with and without an outcome in the shelter (= SIS-category).

In 2006 the RLRR varied between 78 and 87% and increased to 85–93% in 2021 (Fig 11A). One shelter (yellow) had a considerably lower RLRR compared with the others during the first years, but in later years this increased to the same level as the rest. However, more variability in RLRR is seen within shelters than between shelters. From 2006 to 2012 the mean RLRR was comparable with that of 2013 (Fig 11B) and between 2014 and 2021 the mean RLRR increased by 5%.

**Still in shelter (SIS).** Being no part of the outcome, the SIS-category is a specific metric. During the 16 years the SIS remained quite stable, with an overall percentage of 7.1% (SD = 1.7). During the pandemic the SIS dropped to 3.8% in 2020 and increased to 10.1% in 2021. When excluding the pandemic years 2020–2021, the average SIS was still 7.1% (SD = 1.4).

## Discussion

The aims of this study were to apply relevant metrics as indicators of shelter performance during the different phases in shelter care and to identify trends regarding these metrics in seven Dutch shelters during the years 2006 to 2021. The data focused on feline intake per source, outcome per cat, length of stay (LOS) and the risk-based live release rate (RLRR). This study

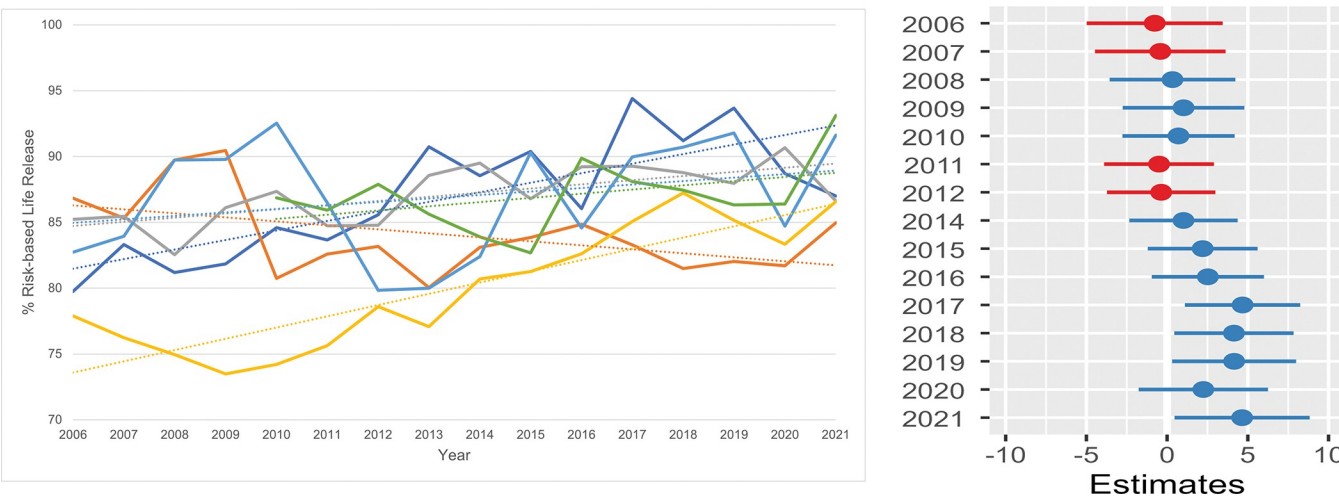

**Fig 11. Risk-based live release rate (RLRR) / shelter. a.** The annual RLRR of all cats per shelter. Five shelters were included in the data from 2006 to 2009 and six shelters from the years 2013 through 2021. Owing to the limited data from shelter G, this shelter was not included in the analysis. **b. Estimates for the average difference in risk-based live release rate (RLRR).** Estimates for the average difference in risk-based live release rate (RLRR) of six shelters compared with the RLRR in 2013. The horizontal lines represent the estimate (the coloured dot) with 95% confidence intervals. An estimate of 0 (zero) means no difference from the annual RLRR in 2013.

showed a significant decrease of almost 40% in the total number of cats per 1000 residents taken in by seven Dutch shelters over the past 16 years. The ratio between the feline sources, however, remained unchanged, with SC being the main source of shelter cats. The major live outcome for shelter cats was being rehomed, but over these 16 years rehoming rates (RR) gradually decreased in most shelters. The second live outcome, return to owner (RTO), was overall increasing. The LOS, especially of the stray cats (SC), showed a decrease over time. The risk-based live release rate (RLRR) gradually increased, caused by a marked decrease in the euthanasia rate, resulting in a moderate increase in live outcomes. The category still-in-shelter (SIS) remained quite stable, except during the COVID pandemic: in 2020, the SIS showed a sharp decline before it peaked in 2021. These results might reflect the unusual situation in shelters during the pandemic. The results of the analysis of the separate shelter metrics for intake, stay and outcome are discussed below.

## Intake

The total intake of shelter cats decreased by a quarter in 16 years, as shelters took in fewer SC and OSC annually. Nothing could be said about the returned adoptions (RetA) as a third source of intake, as the participating shelters used different definitions when taking in these cats, making an analysis impossible. The reduction of intake is not a strict Dutch phenomenon, as Hawes [25] described a decrease of feline intake in 76 shelters and rescue facilities in the state of Colorado (US) of almost 25% during the years 2008–2015. Her next study confirmed this: 225 different animal shelters in Colorado reported a reduction of around 40% during the years 2008–2018 [26]. This contrasted with an Australian study [27] where the annual cat intake in 33 RSPCA shelters during 2006 to 2010, remained the same. However, the Alberthsen study covered a much shorter study period.

The question remains, which factors might be responsible for these internationally observed declines in feline intake? Hawes [26] offered several explanations, such as progress in providing financial and behavioural assistance to keep pets with their families, as well as an increase in social media usage allowing more strays to be returned to their home without the help of

shelters. These aspects could partly explain the decrease in the number of shelter cats in the Netherlands but have never been investigated so far. More obvious explanations, for example a decrease in the number of cats kept as pets, does not hold true for the Netherlands. Although the Dutch pet population has not been monitored systematically and scientifically, based on surveys among pet owners in the period 2018–2021, an increase in the Dutch cat population is reported from 2.6–2.9 to 3.1 million [1]. Another explanation for the decrease in the number of shelter cats could be the success of trap–neuter–return programs (TNR), as TNR activities can reduce the number of (feral) cats and consequently reduce the kitten intake by shelters. Shelters rarely take in adult TNR cats. Only 0.7% of the cats in this study were reported as coming from TNR programs. However young, socialised kittens from TNR programs can be rehomed through shelters. Several studies in different parts of the world have reported a relationship between TNR and feline intake [28–31]. However, this connection has never been verified in the Netherlands, and data from this study are insufficient to substantiate this.

While this decreasing trend in feline intake is seen in some countries, the standardised levels of intake differ internationally. The total feline intake of the Dutch shelters in 2019 (2.6 cats per 1000 residents) appeared to be substantially lower when compared with US data from approximately the same period: Florida: 13–15 cats/1000 residents [28]; Colorado: from 14.1 to 11.6 cats/1000 residents in 2008–2015 [25]; Kentucky: 6.3–3.5 cats/1000 residents in 2011–2019 [29].

Both internationally and in the Netherlands, there is a growing awareness among rescue organisations that well-fed healthy outdoor cats usually do not need shelter care. In some cases, animal ambulances (facilitating the transportation of SC to shelters) might postpone or even cancel these requests for help [32, 33]. This could have an impact on the feline intake, as most of the stray cats are brought to shelters by these ambulances. However, it does not explain the reduction in owner surrendered cats. Owners may have found new ways of rehoming their cats, such as some recent websites offering rehoming services for pets [34, 35], and thus reducing this role of shelters. However, an explanation of these declining feline numbers requires more research and is not fully understood yet.

## Outcome

**Rehoming rate (RR) and return to owner (RTO).** The two major live outcomes were rehoming and RTO. Most cats were rehomed, a process dependent on adequate presentation of the animals to the public [36, 37] and other factors including animal health. Six out of seven shelters showed a gradually decreasing RR. The reasons for this decrease could be caused by a combination of factors, such as the increased RTO. RTO is the second live outcome and can be calculated in different ways: as part of the total intake [29] or related to the number of SC. In this study the RTO is a function of the total stray intake, as recommended by Scarlett [14]. As fewer SC and OSC were taken in by Dutch shelters, while RTO was increasing fewer shelter cats were available for rehoming. An increase in the RTO has a direct dampening effect on the RR. One shelter showed this effect most prominently: with an increasing RTO of 20% of the SC in 2008 to 53% in 2018, its RR decreased from almost 75% in 2009 to 50% in 2018. This shelter explained its RTO increase by an effective use of social media for missing and found pets and being available 7 days a week to allow owners to pick up their stray pets. As Hawes et al. [26] suggested earlier "*whether decreased intake of stray animals by these facilities is associated with increases in a community's use of non-sheltering methods (e.g., social media and mobile applications) to return lost companion animals to their homes*", the data in this study seem to support this assumption.

Following the advice of Scarlett [14], in this study the RTO was calculated as part of the SC (and not as part of the total number of cats taken in) because only stray cats can be returned to their owners. In this way, a shelter can closely monitor its success in returning stray cats to their rightful owner. Shelters in this study showed remarkable high RTO rates when compared internationally. However, this could be caused by differences in definitions. The RTO as a proportion of the total intake (with a larger denominator) is smaller in contrast to the RTO as part of the total stray intake. These differences in the calculation of the RTO make scientific comparison difficult and do not stimulate improvements made by individual shelters. Following Scarlett's lead [14], we therefore recommend calculating the RTO as part of the SC rather than the total feline intake.

**Rates of mortality (MR), euthanasia (ER) and death (DR).** In several publications death rates were determined solely from the group of animals with an outcome (since passing away is an outcome), without acknowledging the risk of dying for every single shelter animal regardless of whether there is an outcome for that animal. The correct way to analyse the mortality in a shelter therefor is by using the MR and ER as these rates not only include cats with an outcome, but also the SIS-part of the population [14, 38]. Earlier European research [16, 17, 39] frequently used the DR instead. To be able to compare the Dutch shelter metrics with this research, the DR was therefore included in this study. Dutch shelters showed a DR of 9.5% (SD = 4.2) of the total feline outcomes during these 16 years. This is significantly lower compared with the DR of 10 Belgium animal shelters in the Brussels area [16]. In 2018 Brussels shelters had DRs of 29%, and in 2019 27%. In our study, only in 2009 did one Dutch shelter report a DR comparable with the Belgian shelters, at 22%. In 2020 the Belgian DR decreased considerable to 11%. In addition, three Czech cat shelters reported DRs of 33% between 2011 and 2015 [17].

The ER of Belgian shelters was also higher compared with the overall average Dutch ER of 7.2% (of the total population): 19 and 15% of Belgian shelter cats were euthanised in 2018 and 2019. The pandemic probably influenced these numbers in Belgium: in 2020 the ER dropped to 4%. The shelter metrics of German shelters are comparable with the Dutch: in 2011 Arhant et al. [39] reported ERs of between 5 and 10% of cats entering German shelters. Following the US methodology, our study standardised euthanasia cases per 1000 human residents. In 2006 the total euthanasia rate of Dutch shelter cats was below 0.60 cats/1000 residents and in 2021 it dropped below 0.20 cats/1000 residents. Although Hawes et al. [25] reports an almost 80% decrease in ERs at shelters in Colorado (from 5.8 cats/1000 residents in 2008 to 1.2 cats/1000 residents in 2015), the Dutch ER is considerably lower. Cultural differences (Europeans prefer an indoor/outdoor life for their felines, while US pet owners prefer their cats to stay indoors [40]), differences in legislation (protection of shelter animals by US legislation is organised at state level [41] and might differ locally [42]), as well as differences in the size of the cat populations (US: 290 cats/1000 residents [40]; the Netherlands: 175 cats/1000 residents) might influence the ER, but more research is needed to explain these differences.

All participating shelters were transparent in providing mortality data on natural deaths and euthanasia of shelter cats. The public perception of good and compassionate care for shelter animals does not always involve euthanasia [43]. When public opinion concerns prompt shelters to under-report this data, it will complicate the analysis and management of potential risk factors for euthanasia [17, 44]. Although it can damage a shelter's reputation, correct reporting of this data is essential.

In individual shelters mortality (MR) and euthanasia rates (ER) can shift with time. When a shelter decreases its ER (an aim for many shelters) but shows no decrease in its MR, this can be caused by a shift from less euthanasia to more natural deaths. When a delay in euthanasia results in more natural deaths, this could be an indication of welfare problems [14], as

euthanasia can shorten the suffering of seriously ill cats. However, in this study we only compared trends in ER without the relation with the MR and further conclusions cannot be made.

**Length of stay (LOS) and return to owner (RTO).** This study showed the relation between the LOS and the RTO cats. RTOs only spent an average of 3.0 days in the shelter, while rehoming the rest of the SC is delayed by the mandatory 14 days of legal holding [3]. Although increasing RTO requires shelter time investment (checking registration of lost pets is labour-intensive), these efforts can reduce the LOS for the total population of SC. Lowering the LOS decreases the risk of cats developing upper respiratory tract disease [11, 45]. The LOS can therefore be a(n) (indirect) tool to monitor feline welfare. When the RTO increases substantially, in shelter metrics it can be useful to indicate the RTO population separately from the total SC population to better monitor the LOS of the rest of the non-RTO-SC.

This steady increase of the RTO, however, might also give a subtle warning. Do these cats, taken in as strays, really need 3 days of shelter facilities before being returned to their owners? Or do we need a new approach to avoid an increased feline circulation through our shelters, with all the risks for the cats involved? Hurley recommends a triage system for shelters at the intake of a stray animal: "*By transporting found animals to a shelter without first making efforts to locate the owner, well-intended finders often unwittingly reduce the likelihood of the pet ever being reunited with its family*" [46]. Although the RTO is rising, the so-called 'stray but well-cared for outdoor cat' risks losing its caretakers when the animal is taken to a shelter, while it could have managed well in its own environment [47].

**Risk-based live release rate (RLRR) and its influencers.** The probability of a Dutch shelter cat being released alive has substantially increased during the past years, to a nine out of ten chance. As the live release in the RLRR (= nominator) consists mainly of two components–RTO and rehoming–it is not affected by a shift between those two outcomes. RTO and rehoming merely act as communicating vessels within the RLRR: the feline outcomes may shift from one to another (more RTO, less rehoming, etc.), but cats still leave the shelter alive. Two other developments, however, substantially impacted the RLRR increase: a decrease in feline intake and a decrease in the ER. The decrease in feline intake could stimulate the rehoming of relatively more elderly and less robust animals (e.g., diabetic or hyperthyroid cats, cats infected with feline retroviruses (FeLV/FIV). When these cats are rehomed instead of euthanised, the RLRR increases. This trend requires improvements in veterinary guidance and animal care and therefore has implications for the level of shelter medicine.

The denominator of the RLRR includes all cats with and without an outcome (the SIS-category). These two parameters can reinforce each other. When cats with a live outcome are leaving the shelter fast (= decrease of the LOS), the SIS decreases, resulting in an increased RLRR. With a lowering of the LOS, the risk of infections decreases as well (for upper respiratory tract disease (URTD) [45]), resulting in a decreasing morbidity and mortality, also resulting in a higher RLRR.

From approximately 120 shelters in Netherlands, a convenience sample of seven shelters was taken which voluntarily participated in this study. Obtaining real data from individual shelters proved to be somewhat challenging. The criterion for participating in this study was the availability of digital shelter data over a time period between 9 and 16 years. Several shelters were therefore unable to join owing to a lack of digital data or too short a time span of the data. A selection bias of this convenience sample can therefore not be excluded. While most shelter metrics were well defined, an information bias was introduced by a variable where each shelter used its own time interval to define returned shelter cats as "returned adoptions". This variable was therefore not included in the analysis. Unfamiliarity with participating in external research projects and sharing the results publicly may also have played a role in some shelters' reluctance to participate. If there had been a central Dutch registration of admission and

outcome of shelter animals, a scientific analysis would probably have been carried out earlier with data from more shelters.

It is important for future research that more shelters participate in comparable studies to be able to verify the observed trends [48], to study existing metrics and develop new statistics for optimising shelter management. Also, the animal species in these studies should not be limited to cats but should include dogs and other species as well. Furthermore, the impact of pet rehoming services and TNR programs for stray or feral cats on the size of shelter populations should be explored.

## Conclusion

This study represents the first long-term analysis of annual Dutch shelter metrics. Trends are presented per individual shelter and from all seven shelters together. The general trends per individual shelter were quite comparable, while their capacities and their locations differed. The presented shelter metrics appear useful for self-monitoring by individual shelters and might help them make decisions for future improvements. These metrics can also be helpful to analyse collective shelter data on a larger scale. The most relevant metrics were Intake (specified by stray cats, surrendered by owner, and others), Outcomes (specified by rehomed, returned to owners, others, and non-live outcomes) and rates such as rehoming rate, return to owner rate, non-live outcome rate, the risk-based live release rate, and the length of stay.

Collaboration between animal shelters is necessary to enable improvement of the quality of care and insight into trends. It is expected that when more Dutch animal shelters systematically provide data for analysis, risk factors for diseases in shelter animals will be better identified, allowing shelters to optimise the health of their animal population and thus monitor and assess the effectiveness of their work. A standardised central registration of the intake and outcome of shelter animals can be developed for this purpose. However, such a registration requires clear definitions per category to enable scientific research. Ultimately, other European shelters could join this shelter data framework and share their knowledge and experiences with each other.

## Supporting information

**S1 File. Residual plots for the studied metrics.** Residual plots of the studied metrics to study the validity of the model about normality and homoscedasticity (constant variability).
(PDF)

**S1 Fig. Annual total intake of cats per shelter.** Annual total intake of cats per shelter. Annual data for seven shelters are used of which five shelters present data during the whole period of 2006–2021, six shelters during 2010–2021 and all seven shelters during 2013–2021. Regardless of shelter size, the total intake of cats decreased between 2006 and 2021 for all shelters in this study, shown also by the decreasing 'Mean / year'.
(TIF)

**S2 Fig. Annual intake of stray cats per shelter.** Annual intake of stray cats per shelter. Annual data for seven shelters are used of which five shelters present data during the whole period of 2006–2021, shelter F during 2010–2021 and shelter G during 2013–2021. Regardless of shelter size, the total intake of stray cats decreased between 2006 and 2021 for all shelters in this study.
(TIF)

**S3 Fig. Death rate (euthanasia and found dead) / shelter.** The annual feline cases of euthanasia and cats found dead as a proportion of the total outcome per shelter is pictured. One shelter had missing information between 2006 and 2009 and another shelter between 2006 and 2012.

All seven shelters were included in the data for the years 2013 through 2021.
(TIF)

**S4 Fig. Estimates for the average difference in death rate (DR) in seven shelters.** Estimates for the average differences in DR in seven shelters compared with 2013. The horizontal lines represent the estimate (the coloured dot) with 95% confidence intervals. An estimate of 0 (zero) means no difference from the annual DR in 2013.
(TIF)

**S5 Fig. Length of stay (LOS) of total cats / shelter.** The median number of annual care days of all incoming cats per shelter is pictured. One shelter had missing information between 2006 and 2009 and another shelter between 2006 and 2012. All seven shelters were included in the data for the years 2013 through 2021. Data supplied by the smallest shelter in this study regarding the LOS (see darkest blue line) from 2020 and 2021 deviated considerably from the metrics of the other shelters. Given its limited size, alterations in the regular shelter management (for example during the SARS-CoV2 pandemic in 2020–2022) could have had a larger impact on its metrics compared with the larger shelters.
(TIF)

**S6 Fig. Risk-based live release.** The annual RLRR of all cats is shown per shelter. One shelter had missing information between 2006 and 2009 and another shelter between 2006 and 2012. All seven shelters were included in the data for the years 2013 through 2021. Data supplied by the smallest shelter in this study regarding the RLRR (see darkest blue line) from 2020 and 2021 deviated considerably from the metrics of the other shelters. Given its limited size, alterations in the regular shelter management (for example during the SARS-CoV2 pandemic in 2020–2022) could have had a larger impact on its metrics compared with the larger shelters.
(TIF)

**S1 Table. Characteristics of participating seven shelters.**
(DOCX)

**S2 Table. The estimated coefficients and their 95% confidence intervals for the intake of shelter cats.**
(DOCX)

**S3 Table. The estimated coefficients and their 95% confidence intervals of the outcome of shelter cats.**
(DOCX)

**S4 Table. The estimated coefficients and their 95% confidence intervals of the LOS and RLRR of shelter cats.**
(DOCX)

## Acknowledgments

The authors would like to thank the Dutch animal shelters for their participation in this project by providing their shelter data for this publication.

## Author Contributions

**Conceptualization:** W. J. R. van der Leij, C. M. Vinke.

**Data curation:** W. J. R. van der Leij, J. C. M. Vernooij.

**Formal analysis:** W. J. R. van der Leij.

**Funding acquisition:** W. J. R. van der Leij.

**Investigation:** W. J. R. van der Leij.

**Methodology:** W. J. R. van der Leij, J. C. M. Vernooij.

**Project administration:** W. J. R. van der Leij.

**Resources:** W. J. R. van der Leij.

**Software:** W. J. R. van der Leij.

**Supervision:** W. J. R. van der Leij, J. C. M. Vernooij, C. M. Vinke, R. J. Corbee, J. W. Hesselink.

**Validation:** W. J. R. van der Leij, J. C. M. Vernooij.

**Visualization:** W. J. R. van der Leij, J. C. M. Vernooij.

**Writing – original draft:** W. J. R. van der Leij.

**Writing – review & editing:** W. J. R. van der Leij, J. C. M. Vernooij, C. M. Vinke, R. J. Corbee, J. W. Hesselink.

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
