## [Decision Letter · Decision Letter 0]

26 Dec 2022

PONE-D-22-32536Quantification of a shelter cat population: trends in intake, length of stay and outcome data of cats in Dutch shelters between 2006 – 2021.PLOS ONE

Dear Dr. der Leij

Thank you for submitting your manuscript to PLOS ONE. After careful consideration, we feel that it has merit but does not fully meet PLOS ONE’s publication criteria as it currently stands. Therefore, we invite you to submit a revised version of the manuscript that addresses the points raised during the review process. The reviewers have raised concerns about your manuscript. Pleal adresses all their pontis in depths and review the manuscript.In addition, the following points should be improved or added to the article:

-sociodemographic and cultural context related to cats in the Netherlands (in the introduction and discussion)..

-implications of the results at the public health level and in the context of One Health.

-presentation of the study area: it must be done in detail and right at the beginning of the methods, together with the locations and characteristics of the shelters included.

-discussion regarding the external validity of the results.

-title: specify that the study refers only to 7 shelters.

-justification for the adoption of the statistical models used, in relation to other alternatives for the analysis of time series.

-presentation of the results of the analysis of residuals.

-titles of figures

We look forward to receiving your revised manuscript.

Kind regards,

Vinícius Silva Belo

Academic Editor

PLOS ONE

Reviewers' comments:

Reviewer's Responses to Questions

**Comments to the Author**

1. Is the manuscript technically sound, and do the data support the conclusions?

Reviewer #1: Yes

Reviewer #2: Yes

2. Has the statistical analysis been performed appropriately and rigorously? 

Reviewer #1: Yes

Reviewer #2: Yes

3. Have the authors made all data underlying the findings in their manuscript fully available?

Reviewer #1: Yes

Reviewer #2: Yes

4. Is the manuscript presented in an intelligible fashion and written in standard English?

Reviewer #1: Yes

Reviewer #2: Yes

5. Review Comments to the Author

Reviewer #1: The study presents the results of original research. The Authors have investigated cat shelter metrics and the trends in the monitored parameters over the period of 16 years. Objectives of the study were clearly defined.

The Introduction is very short and too general. Since the paper analyzes Dutch shelters, the Authors should provide at least some background about the cat shelter operations and relevant legislation in the Netherlands. I find it especially important to clarify if euthanasia of shelter cats for other than health reasons is legal in the Netherlands. And if it is, then whether it was a common practice in the shelters included in the study. The policy concerning euthanasia varies in different countries/shelters and it affects mortality rates greatly.

The methods have been properly described and statistical methods applied allow to make reliable conclusions.

Results are well presented and thoroughly discussed and data interpretation is appropriate.

The Authors described changes in the shelter cat population such as trends in intake, length of stay and outcome data of cats over a long period. Such studies are rare due to lack of consistent data archiving in most animal shelters. Thus, the study presents a significant contribution to the knowledge in this area.

The manuscript is well written, presented and discussed, and understandable to a specialist readership.

Reviewer #2: Line 23 - not sure what is meant by "development of a benchmark" here. Could you reword to clarify? Maybe just moving up some of the language from line 38 to clarify.

Line 33 - if you have a test statistic that justifies the "fewer" bit here, please include it in parentheses

Line 40 - I'm not sure the PLOS guidelines on this, but shouldn't abbreviations go in an appendix? Or be placed inline in the text at first-use?

Line 56 - What about alternate middle steps like foster systems? Or is that still considered "the stay"? If so, please clarify.

Line 69 - it might be useful to briefly touch on the general proportions of animal by species seen in dutch shelters. Cats being the largest population is a fine justification for looking at them, but it could help with context to point out something like "60% of animals in dutch shelter systems are cats, 38% dogs, and 2% other species" or whatever the numbers are

Line 72 - I've noticed you use the term "parameters" frequently throughout this paper - it seems an odd choice of word to me. Parameters tend to indicate some surrounding mathematical or systemic model of which those elements are a component. It seems like the pieces you're describing are more "metrics" or "key performance indicators" (KPIs) than "parameters". It's not a critical point as, at least so far, it's all understandable, but maybe consider this word choice throughout.

Line 79 and 91 - I'm not sure I understand why SIS in included in your Live Release Rate calculation - isn't the "Release" part, by definition, precluding the inclusion of SIS animals? It seems to me it would be appropriate to call out the SIS numbers independently as part of the length of stay measures. You might also just rename the metric to something not including "Release" and that way you can still get the property of signaling for shelter population.

Line 91 - How is Owner Requested Euthanasia handled in this data set?

Line 149 - I'm not sure what "extreme tendencies" means here - could you give more clear information about why the smallest shelter was excluded?

Line 151-160 - Is there a reason you didn't use other variables in the model such as the human populations of the various locations where the shelters are at various time points? A fair bit of evidence suggests that companion animal populations track closely with human populations in an area. I don't know much about dutch population demographics, but this seems an important factor to include in the model. I do see you calculate some values as a proportion of 1000/residents, which is good. I wonder if a more direct inclusion of this factor as a covariate in your model might be appropriate.

S4Fig.tif - The blue line seems to have a huge amount of change around COVID (and generally higher variance) - Why is that one so much more extremely impacted by 2020 than the others? Perhaps this was called out somewhere and I missed it. If the answer was just "COVID", why wouldn't the other shelters be impacted?

Line 469 - Why would you expect a shelter in Colorado to have similar properties to a shelter in the Netherlands? Or is this just coincidence? Are declines in population being driven by similar phenomena in different places where they are occurring?

Line 529 - I'm not sure I've seen a convincing argument this is true. Could you either link to justification or elaborate more on why you believe this? Alternately, you can soften the language.

Line 581 - Although I see the rationale behind this statement, it still needs a citation to back it up. Also, as an aside, I wonder if this might be a reasonable part in the paper to add any citations about ecological effects of free roaming cats (i.e. impact on bird populations, for example). If it is the case that there is a relationship between increases in free roaming cats and declines in bird populations, you might expect the opposite to be true - a decline in free roaming cats leading to an increase in bird populations. If you feel this is out of scope for this section or you don't want to get involved in that area of the literature, that's fine. Just a suggestion.

Line 604 - What sorts of software systems are used in the Netherlands for this? Proprietary ones? Open source ones? Given the push in this paper for standardized, national reporting, you might call out the tools currently being used and any recommended future tool properties you think would facilitate the sorts of reporting for which you are advocating.

Lines 626-635 - starting to get a bit repetitive here, consider condensing this ending to make your point clearly and succinctly

General notes:

Throughout your results, make sure you're including appropriate test statistics and p values for things like linear trends. Also, make sure you're using the appropriate tests for normality of residuals and other checks on those trends.

I appreciate the thoroughness in your reporting of your results. I do think, however, that you might benefit from a more tabular presentation of the key findings. Not every piece of information in the data needs to be reported verbally. Call out the interesting bits in the text, then stick any other potentially informative pieces in a figure/table. Right now, all the content in 177-445 is very verbose. Condensing it might help readers parse the information better.

Your potential interpretations of the declines in the various metrics in your study are good. You might consider adding a few more and/or calling out the possibility that some metrics may be influenced by one phenomena while others may be more associated with a different phenomena. For instance, a reduction in owner surrender cats might be associated with generally improved economic well-being or community education and resources to keep cats in their homes, while a decrease in stray populations might be more associated with an increase in dangerous weather phenomena (total made-up speculation - just illustrating the point that not all factors need be influenced by a single phenomena).

6. PLOS authors have the option to publish the peer review history of their article (what does this mean?). If published, this will include your full peer review and any attached files.

Reviewer #1: No

Reviewer #2: **Yes: **Kevin Horecka

---

## [Author Response · Author response to Decision Letter 0]

8 Mar 2023

Response to reviewers

Concerning: PONE-D-22-32536 - Quantification of a shelter cat population: trends in intake, length of stay and outcome data of cats in seven Dutch shelters between 2006 – 2021.

Journal requirements

1. Journal requirements: Please ensure that your manuscript meets PLOS ONE's style requirements, including those for file naming.

Our answer: our manuscript has been revised according to the PLOS ONE's style requirements, including those for file naming. The use of the Calibri-font in the Figures has been adjusted to the Arial-font.

2. Journal requirements: We note that the grant information you provided in the ‘Funding Information’ and ‘Financial Disclosure’ sections do not match.

Our answer: we did provide the correct information at submission of the revised manuscript.

3. Journal requirements: We note that you have stated that you will provide repository information for your data at acceptance. Should your manuscript be accepted for publication, we will hold it until you provide the relevant accession numbers or DOIs necessary to access your data.

Our answer: when our manuscript will be published, the data will be provided by the repository of DATAVERSE ( https://dataverse.nl/ ). At this moment the data set is in revision to meet the requirements of FAIR data (Findable, Accessible, Interoperable, Reusable). As soon as the data will be available through DATAVERSE, the relevant DOIs necessary to access our data will be provided.

Comments of the academic editor 

The lines referred to are the line numbers in the revised manuscript without track-changes.

Editorial comments: sociodemographic and cultural context related to cats in the Netherlands (in the introduction and discussion).

Our answer: we added to the ‘Introduction’ line 50 - 68: The domestic cat (Felis silvestris catus) is the most popular mammalian pet species in the Netherlands………. Approximately 120 open admission shelters with municipal contracts provide care for stray and owner surrendered cats in the Netherlands.

Editorial comments: implications of the results at the public health level and in the context of One Health.

Our answer: This request caused some confusion because this study does not imply aspects of One Health. In my last study (Serological Screening for Antibodies against SARS-CoV-2 in Dutch Shelter Cats, (2021) Viruses 2021, 13, 1634. https://doi.org/10.3390/v13081634 we did just that: trying to understand the role of the domestic cat in the turmoil of the SARS-CoV-2 pandemic. Based on the results of that study, it seemed unlikely that shelter cats could act as a reservoir of SARS-CoV-2 or pose a (significant) risk to public health. This present study, however, is focussing on the metrics of animal sheltering itself, trying to find the relevant metrics which will enables the shelter medicine world to analyse the leading trends and could be used within shelters for optimalisation of animal care and welfare. 

Editorial comments: presentation of the study area: it must be done in detail and right at the beginning of the methods, together with the locations and characteristics of the shelters included.

Our answer: Information about the seven Dutch animal shelters, is repositioned at the beginning of M&M. However, when providing more information about the locations and characteristics of the shelters, an important aspect should be considered. The participation of these shelters was offered on an anonymity basis because their data could possibly lead to discussion about individual shelters. This was not the intention of this study, and we like to protect the privacy of these shelters. So, we added in line 95 - 97: “A convenience sample of seven local shelters willing to participate in this study under the condition of anonymity, provided annual intake and outcome data of approximately 74,000 individual cats entering and leaving these seven shelters from 2006 through 2021.”

Editorial comments: discussion regarding the external validity of the results.

 Our answer: In this study we selected several key shelter metrics to analyse our own Dutch data. We hope these metrics prove valuable in providing insight into trends in shelters in other countries as well. Our study thus calls on other (European) shelters to share and compare their data in more open and scientific ways, providing opportunities to learn from each other. Only then can the external validity be really substantiated, and can it be determined whether these key metrics could be used in other countries as well. But lack of research haunts us here. In the manuscript itself the external validity is addressed by adding to the discussion line 472 - 487: “The reasons for the declining numbers, as Hawes noted in the Colorado shelters, may or may not also apply to our Dutch shelters ……. However, this connection has never been verified in the Netherlands, and data from this study are insufficient to substantiate this.” 

Editorial comments title: specify that the study refers only to 7 shelters.

Our answer: The title of the manuscript has been changed to: Quantification of a shelter cat population: trends in intake, length of stay and outcome data of cats in seven Dutch shelters between 2006 – 2021.

Editorial comments: justification for the adoption of the statistical models used, in relation to other alternatives for the analysis of time series.

Our answer: we were not sure how to reply on this comment. Could the editor clarify this request?

Editorial comments: presentation of the results of the analysis of residuals.

Our answer: We visually checked the residuals on normality and the constant variance (homoscedasticity) and evaluated each of the models by plotting the residuals as normal probability plot and a scatterplot of predicted values versus residuals.

Editorial comments: titles of figures.

Our answer: our manuscript has been revised according to the PLOS ONE's style requirements, including those for file naming. The use of the Calibri-font in the Figures has been adjusted to the Arial-font.

Review Comments to the Author

Comments Reviewer #1: Since the paper analyses Dutch shelters, the Authors should provide at least some background about the cat shelter operations and relevant legislation in the Netherlands.

Our answer: We added to the ‘Introduction’ line 50 - 66: The domestic cat (Felis silvestris catus) is the most popular mammalian pet species in the Netherlands … Only veterinarians are legally allowed to perform euthanasia.

Comments Reviewer #1: I find it especially important to clarify if euthanasia of shelter cats for other than health reasons is legal in the Netherlands. And if it is, then whether it was a common practice in the shelters included in the study. The policy concerning euthanasia varies in different countries/shelters and it affects mortality rates greatly.

Our answer: We added to the ‘Introduction’ lines 60 - 65: Dutch law only allows euthanasia of cats if an animal poses an immediate danger to humans or animals, when a veterinarian has determined that euthanasia is in the best interest of the animal, or to end the unbearable suffering of the animal… based on the animal's health and well-being.

Comments Reviewer #2: 

Comments Reviewer #2: Line 23 - not sure what is meant by "development of a benchmark" here. Could you reword to clarify? Maybe just moving up some of the language from line 38 to clarify.

Our answer: To clarify, lines 76 – 78 have been added to the ‘Introduction’: “Shelter metrics are essential for the self-assessment of shelters and Dutch animal shelters keep records of the animals in their care, as legally required by stakeholders like municipalities. However, there is also a need for wider use of these metrics to benchmark the progress of shelters.”

Comments Reviewer #2: Line 33 - if you have a test statistic that justifies the "fewer" bit here, please include it in parentheses.

Our answer: We corrected this by:

o deleting: ‘fewer cats’ and replaced this with ‘the number of cats per 1000 residents admitted to Dutch shelters was reduced by 39%’ (line 41).

o the numbers of feline euthanasia decreased and added: ‘with approximately 50%’ (line 42).

Comments Reviewer #2: Line 40 - I'm not sure the PLOS guidelines on this, but shouldn't abbreviations go in an appendix? Or be placed inline in the text at first use?

Our answer: Thank you for this. We deleted the abbreviations according to the PLOS ONE's style requirements.

Comments Reviewer #2: Line 56 - What about alternate middle steps like foster systems? Or is that still considered "the stay"? If so, please clarify.

Our answer: Yes, a shelter stay includes the foster system. We clarified this by mentioning it in three different places in the manuscript: lines 70, 88 and in Table 1.

Comments Reviewer #2: Line 69 - it might be useful to briefly touch on the general proportions of animal by species seen in Dutch shelters. Cats being the largest population is a fine justification for looking at them, but it could help with context to point out something like "60% of animals in Dutch shelter systems are cats, 38% dogs, and 2% other species" or whatever the numbers are.

Our answer: In line 53 we added an estimate of the share of the different animal species in Dutch shelters to the ‘Introduction’: “The animal population in a typical Dutch shelter roughly consists of 70% cats, 20% dogs and 10% other companion animal species.”

Comments Reviewer #2: Line 72 - I've noticed you use the term "parameters" frequently throughout this paper - it seems an odd choice of word to me. Parameters tend to indicate some surrounding mathematical or systemic model of which those elements are a component. It seems like the pieces you're describing are more "metrics" or "key performance indicators" (KPIs) than "parameters". It's not a critical point as, at least so far, it's all understandable, but maybe consider this word choice throughout.

Our answer: thanks for addressing this issue as it is caused by choosing the wrong word in a language that is not our native language. In most cases, we have now replaced the word "parameter" with "metrics".

Comments Reviewer #2: Line 79 and 91 - I'm not sure I understand why SIS in included in your Live Release Rate calculation - isn't the "Release" part, by definition, precluding the inclusion of SIS animals? It seems to me it would be appropriate to call out the SIS numbers independently as part of the length of stay measures. You might also just rename the metric to something not including "Release" and that way you can still get the property of signalling for shelter population.

Our answer: the SIS is used in this ratio to include the animals remaining in the shelter at the end of a period (in this paper at the end of every year). For defining the Release Rate, there are two options: the RR shows the fraction of animals with a live outcome as part of all outcomes (dead or alive) during a certain time. The second option: the RR shows the probability of an animal in the shelter to reach a live outcome as part of the whole animal population present in the shelter during a certain time. Based on the Scarlett’s publication in 2017, we choose for the second option. This somewhat exaggerated example shows why we think this is the better way of presenting these metrics:

- A shelter begins the 1st of August with 100 cats already in their care. During that month 10 more cats enter. On the 31st of August still 90 cats are still in the shelter (the SIS portion), while 20 cats reached an outcome: 10 are adopted, 3 returned to their owner and 7 euthanised. What is the best way to calculate the adoption rate?

10 of 20 cats with final outcome = 50% adoption rate?

10 of 10 in august entering cats = 100% adoption rate? 

10 of the total 110 cats being cared for in this shelter (including the SIS portion) = (10/(20+90)*100 during the month of august = 9,1% adoption rate.

- The third way of calculating represent the chance of every individual cat during its shelter time to have a live release (the proportion of cats in the shelter ‘at risk of adoption’). Scarlett strongly recommends this way of calculating an outcome as part of all the cats available in the shelter, not only the cats which reach an outcome (Scarlett JM, Greenberg M, Hoshizaki T. Every nose counts. Using Metrics in Animal Shelters. 1st ed. A Maddie’s Guide. Maddie’s Fund; 2017. Chapter 4 page 50 -51, 58). 

Comments Reviewer #2: Line 91 – Q: How is Owner Requested Euthanasia handled in this data set?

Our answer: we added to the ‘Introduction’ and the M&M:

- line 63: “Dutch shelters do not facilitate owners' requests for euthanasia of their pet, as this assessment is invariably made by the shelter itself based on the animal's health and well-being.”

- Line 136: “Euthanasia at the owner's request does not play a role in this study, because this practice is unusual for Dutch animal shelters.”

Comments Reviewer #2: Line 149 - I'm not sure what "extreme tendencies" means here - could you give more clear information about why the smallest shelter was excluded?

Our answer: we rephrased “extreme tendencies” with ‘irregular’ in line 133: “The smallest shelter (S1 Fig, darkest blue line) in this study showed an irregular annual LOS (S4 Fig) and risk-based live release rate (RLRR) (S5 Fig).” 

Comments Reviewer #2: Line 151-160 - Is there a reason you didn't use other variables in the model such as the human populations of the various locations where the shelters are at various time points? A fair bit of evidence suggests that companion animal populations track closely with human populations in an area. I don't know much about Dutch population demographics, but this seems an important factor to include in the model. I do see you calculate some values as a proportion of 1000/residents, which is good. I wonder if a more direct inclusion of this factor as a covariate in your model might be appropriate.

Our answer: The metrics per 1000 residents in the shelter care area were used for normalisation of these metrics (I therefor corrected the omitting 'per 1000 residents' in the ‘statistics’ chapter, lines 157). But instead of distinguishing between individual shelters, this specific study focused on shelter metrics that showed a common trend when used for different shelters. For readers however, who want more detailed information, the S1 table was added in which the average annual income per resident in the shelter care area (data from 2019) and the degree of urbanization of the shelter care area are given per shelter. 

Comments Reviewer #2: S4Fig.tif – The blue line seems to have a huge amount of change around COVID (and generally higher variance) - Why is that one so much more extremely impacted by 2020 than the others? Perhaps this was called out somewhere and I missed it. If the answer was just "COVID", why wouldn't the other shelters be impacted?

Our answer: we added lines 161- 166: “Data supplied by the smallest shelter in this study regarding the LOS and the RLRR from 2020 and 2021 deviated considerably (see S4 and S5 Fig, darkest blue line) from the metrics of the other shelters. Because of its limited size, alterations in the regular shelter management (like during the SARS-CoV2 pandemic in 2020 – 2022) could have had more impact on its metrics compared with the larger shelters. We therefor excluded the LOS and RLRR of this small shelter data from the general analysis.” 

This shelter is the smallest of all shelters in this study. The pandemic could have had a relatively larger effect on this shelter because of its limited size (= less employees and volunteers). Our study included 2 years of the pandemic, but the effect of the pandemic was not a part of the research aim. If that would have been the case, our study design would have been different.

Comments Reviewer #2: Line 469 - Why would you expect a shelter in Colorado to have similar properties to a shelter in the Netherlands? Or is this just coincidence? Are declines in population being driven by similar phenomena in different places where they are occurring?

Our answer: In this study we selected several key shelter metrics to analyse our own Dutch data. We hope these metrics prove valuable in providing insight into trends in shelters in other countries as well. Our study thus calls on other (European) shelters to share and compare their data in more open and scientific ways, providing opportunities to learn from each other. Only then can the external validity be really substantiated, and can it be determined whether these key metrics could be used in other countries as well. But lack of research haunts us here. In the manuscript itself the external validity is addressed by adding to the discussion line 472 - 487: “The reasons for the declining numbers, as Hawes noted in the Colorado shelters, may or may not also apply to our Dutch shelters ……. However, this connection has never been verified in the Netherlands, and data from this study are insufficient to substantiate this.” 

Comments Reviewer #2: Line 529 – “The correct way to analyse the mortality in a shelter is by using the MR and ER, as these rates not only include cats with an outcome, but also the SIS-part of the population.” I'm not sure I've seen a convincing argument this is true. Could you either link to justification or elaborate more on why you believe this? Alternately, you can soften the language.

Our answer: Janet M. Scarlett, co-author of the [48] paper and writer of the book ‘Every nose counts, Using metrics in Animal shelters’, stated on page 57: “The mortality (or death) rate includes natural deaths in a time period divided by outcome-eligible animals in that time period.” She defines outcome-eligible animals as “animals in the shelter at the very beginning of the interval” (pg 53), being animals with and without an outcome yet. Scarlett intended to interchange ‘mortality rate’ with ‘death rate’. Unfortunately, many authors defined their euthanasia or death rates as a section of animals exclusively with an outcome [32,33,34], ignoring the animals still in the shelter in risk of dying. In this paper the Mortality and Euthanasia rate include the SIS portion in the denominator, while Death rate has only the animals with an outcome in the denominator, according to Scarlett [31].

Comments Reviewer #2: Line 581 - Although I see the rationale behind this statement, it still needs a citation to back it up. Also, as an aside, I wonder if this might be a reasonable part in the paper to add any citations about ecological effects of free roaming cats (i.e. impact on bird populations, for example). If it is the case that there is a relationship between increases in free roaming cats and declines in bird populations, you might expect the opposite to be true - a decline in free roaming cats leading to an increase in bird populations. If you feel this is out of scope for this section or you don't want to get involved in that area of the literature, that's fine. Just a suggestion. 

Our answer: a link to the report of Mrs. Marcia Mayeda, was added to underline our observations. (Report Back on Managed Intake and Best Practices Within DACC Care Centers. Los Angeles, CA: Los Angeles Department of Animal Care and Control. 2021. Available from: https://www.documentcloud.org/documents/21048959-la-county-report-back-animal-control ). However, we believe that the relationship between free-ranging cats and their prey population is a complicated and multifactorial relationship and beyond the scope of this study.

Comments Reviewer #2: Line 604 - What sorts of software systems are used in the Netherlands for this? Proprietary ones? Open-source ones? Given the push in this paper for standardized, national reporting, you might call out the tools currently being used and any recommended future tool properties you think would facilitate the sorts of reporting for which you are advocating.

Our answer: The two software programs used in Dutch shelters are commercially available. If a standardized central registration of shelter animals is set up in the future, it will probably be a government initiative. If these commercial software developers play a role in this, we will point in lines 635 – 638 of the ‘Conclusions’: “A standardized central registration of the intake and outcome of shelter animals can be developed for this purpose. However, such a registration requires clear definitions per category to enable scientific research.” 

Comments Reviewer #2: Lines 626-635 - starting to get a bit repetitive here, consider condensing this ending to make your point clearly and succinctly.

Our answer: Our excuses for this error in the text: there has been an accidental replication in the text, which has been removed.

General notes:

Comments Reviewer #2: Throughout your results, make sure you're including appropriate test statistics and p values for things like linear trends. 

Our answer: Regular regression lines were added to the Figures 1a – 11a, as a visual representation of the trends in the data, but these lines were not used for statistical analysis.

Comments Reviewer #2: Also, make sure you're using the appropriate tests for normality of residuals and other checks on those trends.

Our answer: We visually checked the residuals on normality and the constant variance (homoscedasticity) and evaluated each of the models by plotting the residuals as normal probability plot and a scatterplot of predicted values versus residuals.

Comments Reviewer #2: I appreciate the thoroughness in your reporting of your results. I do think, however, that you might benefit from a more tabular presentation of the key findings. Not every piece of information in the data needs to be reported verbally. Call out the interesting bits in the text, then stick any other potentially informative pieces in a figure/table. Right now, all the content in 177-445 is very verbose. Condensing it might help readers parse the information better.

Our answer: Thank you for this comment: by considerably shortening the text of the 'Results', we believe that the readability of this chapter has improved.

Comments Reviewer #2: Your potential interpretations of the declines in the various metrics in your study are good. You might consider adding a few more and/or calling out the possibility that some metrics may be influenced by one phenomena while others may be more associated with a different phenomena. For instance, a reduction in owner surrender cats might be associated with generally improved economic well-being or community education and resources to keep cats in their homes, while a decrease in stray populations might be more associated with an increase in dangerous weather phenomena (total made-up speculation - just illustrating the point that not all factors need be influenced by a single phenomena).

Our answer: The reasons for the declining numbers, as Hawes noted in the Colorado shelters, may or may not also apply to our Dutch shelters. Lack of research haunts us here. That's why we added lines 472 – 487 to the 'Discussion': “Which factors might be responsible for these internationally observed declines in feline intake? ……. However, this connection has never been verified in the Netherlands, and data from this study are insufficient to substantiate this.”

Sincerely yours

Ruth van der Leij (corresponding author)

---

## [Decision Letter · Decision Letter 1]

12 Apr 2023

PONE-D-22-32536R1Quantification of a shelter cat population: trends in intake, length of stay and outcome data of cats in seven Dutch shelters between 2006 – 2021.PLOS ONE

Dear Dr. van der Leij,

Thank you for submitting your manuscript to PLOS ONE. After careful consideration, we feel that it has merit but does not fully meet PLOS ONE’s publication criteria as it currently stands. Therefore, we invite you to submit a revised version of the manuscript that addresses the points raised during the review process. I would like to express my gratitude for the effort put into revising the article. However, as requested by reviewer 2, please consider having the text proofread by a native speaker, as the language quality needs improvement. This will ensure that the article meets the publication standards. Additionally, please include the graphics of the residual analyses as supplementary material. Regarding the question about the statistical analysis, I apologize for not being clear enough in my previous message. I only ask that you provide a brief explanation of why the linear mixed model was chosen and its properties. This will be useful for readers who are less familiar with the analysis.

We look forward to receiving your revised manuscript.

Kind regards,

Vinícius Silva Belo

Academic Editor

PLOS ONE

Journal Requirements:

Reviewers' comments:

Reviewer's Responses to Questions

**Comments to the Author**

1. If the authors have adequately addressed your comments raised in a previous round of review and you feel that this manuscript is now acceptable for publication, you may indicate that here to bypass the “Comments to the Author” section, enter your conflict of interest statement in the “Confidential to Editor” section, and submit your "Accept" recommendation.

Reviewer #1: All comments have been addressed

Reviewer #2: All comments have been addressed

2. Is the manuscript technically sound, and do the data support the conclusions?

Reviewer #1: Yes

Reviewer #2: Yes

3. Has the statistical analysis been performed appropriately and rigorously? 

Reviewer #1: Yes

Reviewer #2: Yes

4. Have the authors made all data underlying the findings in their manuscript fully available?

Reviewer #1: Yes

Reviewer #2: Yes

5. Is the manuscript presented in an intelligible fashion and written in standard English?

Reviewer #1: Yes

Reviewer #2: Yes

6. Review Comments to the Author

Reviewer #1: All comments have been addresed, the authors have revised the manusript accordingly. I recommend to accept the manuscript for publication. However, the manuscript would benefit from a thorough proof-reading by a native speaker, the language quality needs to be improved.

Reviewer #2: Thank you for addressing all of my comments. I believe you did so effectively and in a way which greatly improved the paper. I have no more concerns at this time that I believe can be directly addressed in this work. Although I'm not entirely convinced I understand the strengths and weaknesses of the specific metrics you're using here, I agree with you that more studies using these metrics should help us better identify ways in which the metrics can be valuable for shelters (and if they need to be modified in any way). Great work!

7. PLOS authors have the option to publish the peer review history of their article (what does this mean?). If published, this will include your full peer review and any attached files.

Reviewer #1: No

Reviewer #2: **Yes: **Kevin Horecka

---

## [Author Response · Author response to Decision Letter 1]

1 May 2023

Response to Reviewers

Concerning: PONE-D-22-32536R1 - Quantification of a shelter cat population: trends in intake, length of stay and outcome data of cats in seven Dutch shelters between 2006 and 2021.

The lines referred to are the line numbers in the revised manuscript without track-changes.

Comments of the academic editor:

1- Editorial comments: As requested by reviewer 2, please consider having the text proofread by a native speaker, as the language quality needs improvement. This will ensure that the article meets the publication standards.

Our answer: the text was proofread and corrected by a native speaker.

2- Editorial comments: please include the graphics of the residual analyses as supplementary material.

Our answer: we added the graphics of the residual analyses as supplementary material as a S1 File: Residual plots for the studied metrics. Residual plots of the studied metrics to study the validity of the model about normality and homoscedasticity (constant variability).

3- Editorial comments: provide a brief explanation of why the linear mixed model was chosen and its properties. This will be useful for readers who are less familiar with the analysis. 

Our answer: we added to the chapter ‘Statistics’ line 167 - 181: 

For each shelter, the metrics were standardized and calculated per year. The metrics were repeatedly calculated and therefore correlated within the shelter. Although the metrics were standardized, the level of the metrics i.e., “CATS IN per year” for a specific shelter can be affected by its shelter management, finances etc. Simple linear regression models are based on linearity between outcome variable and continuous independent variable, normal distribution, and homoscedasticity (constant variability) of the residuals of independent observations. When the independent variable is categorical then this is called ANOVA and belongs to the family of linear regression models. The assumption of independence is violated in our data collection and the dependency between measurements of the same metric within shelter should be incorporated in the statistical model. We therefore used a linear mixed effects model which is a combination of estimation of fixed effects and so-called random effects i.e., estimation of variability between the independent subjects (shelter) as metrics were measured yearly within the same shelter. For comparison a very simple example of a model taking dependency of measurements into account is the paired t-test with two conditions within the same subject (i.e., body weight before and after diet treatment of the same person).

---

## [Editor Report · Decision Letter 2]

5 May 2023

Quantification of a shelter cat population: trends in intake, length of stay and outcome data of cats in seven Dutch shelters between 2006 and 2021.

PONE-D-22-32536R2

Dear Dr. vad der Leji,

We’re pleased to inform you that your manuscript has been judged scientifically suitable for publication and will be formally accepted for publication once it meets all outstanding technical requirements.

Kind regards,

Vinícius Silva Belo

Academic Editor

PLOS ONE
---

## [Editor Report · Acceptance letter]

10 May 2023

PONE-D-22-32536R2 

Quantification of a shelter cat population: trends in intake, length of stay and outcome data of cats in seven Dutch shelters between 2006 and 2021. 

Dear Dr. van der Leij:

I'm pleased to inform you that your manuscript has been deemed suitable for publication in PLOS ONE. Congratulations! Your manuscript is now with our production department. 

Kind regards, 

on behalf of

Dr. Vinícius Silva Belo 

Academic Editor

PLOS ONE